# ReGuidance: Diffusion Steering with Strong Latent Initializations Solves Hard Inverse Problems

## Abstract

In recent years there has been a flurry of activity around using pretrained diffusion models as informed data priors for solving inverse problems, and more generally around steering these models towards certain reward models. Training-free methods like gradient guidance have offered simple, flexible approaches for these tasks, but when the reward is not informative enough, e.g., in inverse problems with highly compressive measurements, these techniques can veer off the data manifold, failing to produce realistic data samples. To address this challenge, we devise a simple algorithm, **ReGuidance**, that leverages prior methods' solutions as strong initializations and substantially enhancing their realism. Given a candidate solution $x$ produced by a given method, we propose inverting the solution by running the unconditional probability flow ODE in reverse starting from $x$, and then using the resulting latent as an initialization for a simple instantiation of diffusion guidance. In toy settings, we provide theoretical justification for why this technique boosts the reward and brings $x$ closer to the data manifold. Empirically, we evaluate our algorithm on difficult image restoration tasks including large box inpainting, heavily downscaled superresolution, and high noise deblurring with both linear and nonlinear blurring operations. We find that, using a wide range of baseline methods as initializations, applying our method results in much stronger samples with better realism and measurement consistency.

## 1 Introduction

Motivated by the flexibility and fidelity with which diffusion models can capture realistic data distributions Ho et al. (2020); Dhariwal & Nichol (2021); Song et al. (2021), a large number of recent works have sought to leverage these models as rich data priors for solving complex downstream tasks like Bayesian inference problems Baldassari et al. (2023); Venkatraman et al. (2024); Chan et al. (2025), black box optimization Krishnamoorthy et al. (2023); Li et al. (2024c), medical imaging Chung & Ye (2022); Chung et al. (2022a); Dorjsembe et al. (2024); Hung et al. (2023), and molecular design Gruver et al. (2023); Wohlwend et al. (2024). These tasks are all incarnations of the general problem of *reward guidance*: given a pretrained model for data distribution $q$ and reward model $r$, design a procedure that generates samples $x$ which simultaneously are "*realistic*", in that they have high likelihood under $q$, and achieve high *reward $r(x)$*.

Despite significant strides in practice along this direction, our understanding of this task remains limited both mathematically and empirically. It is common to frame reward guidance as sampling from the tilted density $\tilde{q}(x) \propto q(x) \cdot e^{r(x)}$. However, it is very unclear to what extent methods in practice are actually accomplishing this, as they either rely on heuristic approximations that significantly bias the output away from sampling from $\tilde{q}$ Chung et al. (2023); Kawar et al. (2022); Zhang et al. (2024), or they rely on stochastic optimal control, for which computational costs prevent training for long enough to actually approach $\tilde{q}$ Denker et al. (2024); Domingo-Enrich et al. (2025). Indeed, for certain simple choices of $q$ and $r$, prior works have even shown that sampling from the tilted density is *computationally intractable* Gupta et al. (2024); Bruna & Han (2024).

**Hard reward models.** While this seems to run counter to the impressive capabilities of guidance methods in practice, it is not difficult to construct natural reward models under which existing

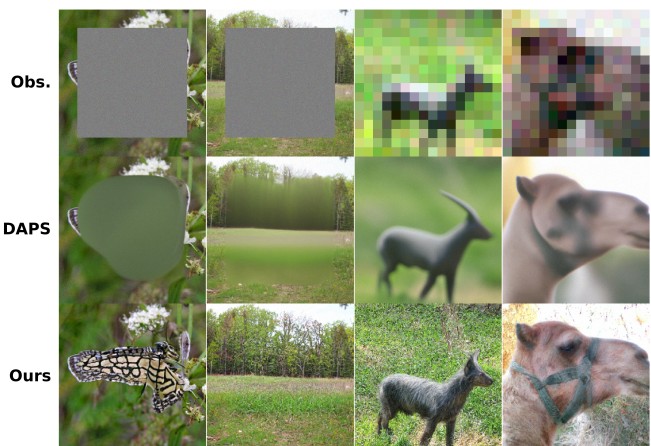

**Figure 1:** Comparing restoration performance on hard box-inpainting (cols. 1,2) and superresolution (cols. 3,4). First row is the measurement, second gives solutions given by state-of-the-art DAPS method Zhang et al. (2024), and third gives solutions obtained by applying our method REGUIDANCE to DAPS.

methods fail. In this work, we focus on rewards arising from *hard inverse problems* with highly compressive measurements. A canonical example is *large box-inpainting*, where an $m \times n$ image is observed with a random region of size $\alpha m \times \alpha n$ masked out, for some large masking fraction $\alpha$. The reward function $r(x)$ in this case is given by the negative squared distance between the generated sample $x$ and the observed masked image, restricted to the unmasked pixels.

State-of-the-art methods for reward guidance solve this problem remarkably well for data distributions like face datasets, which have large amounts of redundancy and low intrinsic dimension. But as soon as one moves beyond these settings, these methods break down even for moderate $\alpha$ (e.g. $\alpha > 1/2$). Indeed, as shown in Figure 1 for hard inpainting and superresolution, these methods not only fail to sample from the appropriate tilted density (i.e., the posterior) in such regimes, but their outputs visibly suffer from poor realism. On the other hand, it is not so hard to achieve high reward: the methods pictured all generate outputs consistent with the given measurements.

Altogether this basic example suggests that while it is now par for the course to be able to generate highly realistic and accurate reconstructions for "easy" inverse problems, current methods are just not there yet when it comes to these more challenging reward models.

### 1.1 OUR CONTRIBUTIONS

Our main results are threefold:

**1. A new and simple boosting method:** We propose **REGUIDANCE**, a simple algorithm that leverages strong latent initializations in conjunction with diffusion guidance. It takes as input a candidate reconstruction $x$, produced by an inverse problem solver of the user's choice, and generates a new reconstruction. The algorithm is simple, operating in two modular steps (see Algorithm 1 for the pseudocode). **(I)** It runs the deterministic sampler given by the pretrained diffusion model for the base density $q$ *in reverse* to extract the latent $x^*$ associated to $x$. **(II)** Starting from $x^*$, it runs the *Diffusion Posterior Sampling (DPS)* algorithm of Chung et al. (2023) to produce the new reconstruction $x^{\mathsf{DPS}}$. We provide a full description of the technical details in Section 3.

We show that in many settings, both empirical and theoretical, the new reconstruction achieves higher realism while still achieving high reward.

**2. Empirical performance:** In traditionally difficult inverse tasks for image restoration, we show that REGUIDANCE can significantly boost the measurement consistency and sample quality of candidate reconstructions. For inverse problems like *box-inpainting*, *superresolution*, and *deblurring*, we observe a consistent and significant improvement in both reward and realism metrics when applying REGUIDANCE. Qualitatively, we also observe diverse but realistic reconstructions that are meaningfully distinct from the original sample being measured (details in Section 4).

Our work investigates a largely unexplored design axis, i.e. the choice of latent initialization affects realism, and shows that selecting a good latent can lead to substantial performance improvements, providing solutions to inverse tasks that prior sampling methods tend to collapse on.

**3. Theoretical justification:** For two models of Gaussian mixture data, we rigorously prove that REGUIDANCE improves reward value and realism. Formally, in these settings we are able to show that the algorithm approximately brings $x$ onto the manifold of points which achieve maximal reward, and that even if $x$ already lies on this manifold, under REGUIDANCE it will get contracted even closer to one of the modes of the data. These results are given in Sections 3.1 and 3.2.

To our knowledge, these are the first end-to-end theoretical guarantees for DPS in any model of data. This is enabled by two shifts in perspective. First, instead of sampling from the tilt $\tilde{q}$, which DPS provably fails to do even on simple models of data (Appendix B), we focus on the goal of producing a sample with high likelihood under the base density ("realism") and high measurement consistency ("reward"). Second, because we are quantifying how much our algorithm "improves" $x$, our bounds are inherently tied to the choice of latent $x^*$ that DPS is initialized at in Step (II) of REGUIDANCE. One corollary of our results is DPS's performance depends heavily on the quality of the initial latent.

## 1.2 RELATED WORK

**Solving inverse problems with pretrained diffusion models.** Pretrained generative models like GANs and diffusion models have been extensively used as data priors for solving inverse problems Bora et al. (2017); Ongie et al. (2020); Daras et al. (2024). In recent years, the diffusion models have outperformed GANs at modeling the image prior and generating novel images, leading to increased interest in using diffusion models to solve inverse problems Lugmayr et al. (2022); Kawar et al. (2022); Saharia et al. (2023); Chung et al. (2023); Wang et al. (2023); Song et al. (2023); Zhu et al. (2023); Zhang et al. (2024); Li et al. (2024a); MOUFAD et al. (2025); Domingo-Enrich et al. (2025); Chen et al. (2025). Many *training-free* methods such as DDRM Kawar et al. (2022), DPS Chung et al. (2023), ΠGDM Song et al. (2023), DAPS Zhang et al. (2024) and *training-based* methods Denker et al. (2024); Domingo-Enrich et al. (2025) have been proposed. Training-free methods modify the reverse process of diffusion models with a hand-designed guidance term that pushes the trajectory towards measurement consistency. Our work provides a simple method to boost the performance of such methods for inverse problems where the measurement is highly lossy.

**Reward guidance for generative models.** A closely related but more general line of work concerns steering the outputs of pretrained models to generate samples from a tilt given by a reward model Black et al. (2024); Fan et al. (2023); Wallace et al. (2023); Clark et al. (2024); Domingo-Enrich et al. (2025). When the reward is given by measurement consistency, the problem reduces to posterior sampling for inverse problems. While our method in principle can also be applied to this more general setting, in this work we focus primarily on image restoration tasks.

**Scaling inference-time compute for diffusion models.** Recent works have shown that increasing inference-time compute for diffusion models can improve performance across various generation tasks Dou & Song (2024); Wu et al. (2024); Li et al. (2024b); Uehara et al. (2025); Singhal et al. (2025); Ma et al. (2025). These methods typically start sampling multiple diffusion generation trajectories called particles, and reweight and filter the particles during the generation to optimize for the reward. Reweighting and filtering of the particles is performed using sequential Monte Carlo guidance Dou & Song (2024); Wu et al. (2024) or value-based importance sampling Li et al. (2024b). Our work provides a different way of scaling inference-time compute by selecting a good latent noise vector by running the reverse unconditional probability flow ODE.

## 2 TECHNICAL PRELIMINARIES

### 2.1 DIFFUSION MODEL BASICS

In the context of *unconditional* generation, diffusion models provide the following framework for approximately sampling from a target measure $q$ over $\mathbb{R}^d$ given access to samples from $q$. In this work we work with the most common choice of forward process, the *Ornstein-Uhlenbeck process*, which is given by the SDE $\mathrm{d}x_t = -x_t\,\mathrm{d}t + \sqrt{2}\mathrm{d}B_t$, where $(B_t)_{t\geq 0}$ denotes a standard Brownian motion in $\mathbb{R}^d$ and $x_0 \sim q$. Define $q_t \triangleq \mathrm{law}(x_t)$. Given a large terminal time $T \geq 0$, one choice of SDE which provides a time-reversal for this process over times $t \in [0, T]$ is the *reverse SDE*

$$\mathrm{d}x_t^{\leftarrow} = (x_t^{\leftarrow} + 2\nabla \ln q_{T-t}(x_t^{\leftarrow}))\,\mathrm{d}t + \sqrt{2}\mathrm{d}B_t\,,$$

where now $(B_t)_{0 \leq t \leq T}$ denotes the reversed Brownian motion, and the *score functions* $(\nabla \ln q_t)_t$ are estimated from data. We will touch upon issues of estimation error in Sections 3.1 and 3.2, but for now we will assume these are exactly known to us.

The reverse SDE has the property that if $x_T^{\leftarrow} \sim \mathcal{N}(0, \mathrm{Id})$, then $\mathrm{law}(x_t^{\leftarrow}) \approx q_{T-t}$ for all $0 \leq t \leq T$ provided $T$ is sufficiently large. In particular, if one can simulate the reverse SDE up to $t = T$, then the resulting iterate is distributed as a sample from the target measure.

Another process which also yields a time-reversal of the forward process is the *probability flow ODE*

$$\mathrm{d}x_t^{\leftarrow} = (x_t^{\leftarrow} + \nabla \ln q_{T-t}(x_t^{\leftarrow})) \, \mathrm{d}t \,.$$

This is the process that is used in denoising diffusion implicit models (DDIMs) and, equivalently, flow matching models with Gaussian source distribution.

## 2.2 POSTERIOR SAMPLING WITH DIFFUSION MODELS

In this work we are interested in the general question of *steering* a diffusion model according to a given reward model. In this setting, one is given access to the scores $\nabla \ln q_t$ for a base measure $q$, corresponding to a *pretrained* diffusion model, as well as access to a *reward model* $r : \mathbb{R}^d \to \mathbb{R}$, and the goal is to design a sampler for the tilted measure $\tilde{q}(x) \propto q(x) \cdot e^{r(x)}$. Various training-free methods Kawar et al. (2022); Chung et al. (2023); Zhang et al. (2024) have been proposed for sampling from the tilt, and our work offers a cheap training-free boosting method.

A well-studied family of reward models is those arising from *inverse problems*. Suppose that a signal $x$ is sampled from the base measure $q$, and we observe $y = f(x) + g$, where $f : \mathbb{R}^d \to \mathbb{R}^m$ and $g \sim \mathcal{N}(0, \sigma^2 \mathrm{Id})$. Then conditioned on observing $y$, the posterior measure on $x$ is given by

$$\tilde{q}(x) \propto q(x) \cdot e^{r(x)} \,, \qquad\qquad r(x) = -\frac{1}{2\sigma^2} \|y - f(x)\|^2 \,.$$

We will often refer to $\|y - f(x)\|^2$ as the *reconstruction loss*.

One of the most popular training-free approaches for trying to sample from this posterior is *diffusion posterior sampling (DPS)* Chung et al. (2023). First, one notes that

$$\nabla \ln \tilde{q}_t(x) = \nabla \ln q_t(x) + \nabla \ln \mathbb{E}_{x_0}[e^{r(x_0)} \mid x_t = x] \,,$$

where the conditional expectation is with respect to $x_0$ conditioned on $x_t \sim \mathcal{N}(e^{-t}x_0, (1 - e^{-2t})\mathrm{Id})$ being equal to $x$. Unfortunately, this vector field is not readily available as it requires gradients of the posterior density on $x_0$, which can be very complicated. DPS offers one popular heuristic: replace the expectation with the point mass at $\mu_t(x) \triangleq \mathbb{E}[x_0 \mid x_t = x]$. This results in the approximation

$$\nabla \ln \tilde{q}_t(x) \overset{?}{\approx} \nabla \ln q_t(x) + v_t^{\mathsf{DPS}}(x) \qquad\qquad v_t^{\mathsf{DPS}}(x) \triangleq \nabla_x r(\mathbb{E}[x_0 \mid x_t = x]) \,.$$

One can then try sampling from $\tilde{q}$ by running either the ODE

$$\mathrm{d}x_t^{\mathsf{DPS}} = (x_t^{\mathsf{DPS}} + \nabla \ln q_{T-t}(x_t^{\mathsf{DPS}}) + v_{T-t}^{\mathsf{DPS}}(x_t^{\mathsf{DPS}}))\mathrm{d}t \qquad\qquad \text{(DPS-ODE)}$$

or the analogous SDE, starting from $x_0^{\mathsf{DPS}} \sim \mathcal{N}(0, \mathrm{Id})$.

In this work, we will focus on *linear* inverse problems for concreteness, in which case $f(x) = Ax$ for $A \in \mathbb{R}^{m \times d}$. For linear inverse problems, we have

$$v_t^{\mathsf{DPS}}(x) = \nabla_x r(\mathbb{E}[x_0 \mid x_t = x]) = \frac{1}{\sigma^2} \nabla \mu_t(x) A^{\top}(y - A\mu_t(x)) \,,$$

where $\nabla \mu_t \in \mathbb{R}^{d \times d}$ denotes the Jacobian of the denoiser.

Unfortunately, it is well-known that DPS incurs significant bias relative to the true reverse process for $\tilde{q}$, even for very simple special cases like $q = \mathcal{N}(0, \mathrm{Id})$ and $A = \mathrm{Id}$ (see Appendix B). Indeed, to our knowledge there have been no works providing a well-defined, theoretically rigorous guarantee for what DPS is actually accomplishing, despite its surprising effectiveness in practice. Nevertheless, in this work we show that by starting this process with the appropriate initialization for $x_0^{\mathsf{DPS}}$, we can precisely pin down where Eq. (DPS-ODE) ends up after time $T$.

## 3    THE REGUIDANCE ALGORITHM AND THEORETICAL GUARANTEES

Here we give a complete description of our algorithm. Suppose we get as input initial reconstruction $x$, generated by an algorithm of the user's choice and intended to be an approximate sample from the support of $q$ that achieves some decent level of reward $r(x)$. Our algorithm consists of two simple steps, given in Algorithm 1 below.

---

**Algorithm 1:** REGUIDANCE$(x, r)$

---

**Input**          : Initial reconstruction $x \in \mathbb{R}^d$ and a reward model $r : \mathbb{R}^d \to \mathbb{R}$
                 /* We set $r(x) = \|y - Ax\|^2$ for inverse tasks          */
**Hyperparams:** Guidance strength $\rho$, time horizon $T$ for ODEs
**Output**        : Improved reconstruction $\hat{x}$

1 **Extract latent:** Run the unconditional probability flow ODE *in reverse* from the initial
   reconstruction $x$ to obtain latent $x_T^*$:

$$\mathrm{d}x_t^* = -(x_t^* + \nabla \ln q_t(x_t^*)) \, \mathrm{d}t, \qquad x_0^* = x$$

2 **Run DPS from latent:** Run the DPS-ODE for time $T$ starting from $x_T^*$:

$$\mathrm{d}x_t^{\mathsf{DPS}} = \left(x_t^{\mathsf{DPS}} + \nabla \ln q_{T-t}(x_t^{\mathsf{DPS}}) + \rho \nabla_x r(\mu_{T-t}(x_t^{\mathsf{DPS}}))\right) \mathrm{d}t, \quad x_0^{\mathsf{DPS}} = x_T^*$$

   where $\mu_{T-t}(x) \triangleq \mathbb{E}[x_0 \mid x_{T-t} = x]$.
3 **return** $\hat{x} = x_T^{\mathsf{DPS}}$.

---

As we will see in our analysis (see Theorem 2) and in experiments (see Section 4.3), it is crucial that the second step in REGUIDANCE uses the *ODE* formulation of DPS rather than the SDE. And as the next two sections will make clear, it is also essential that we initialize DPS at the latent $x_T^*$ rather than simply at some random latent as in the standard implementation of DPS.

### 3.1    THEORY VIGNETTE 1: BOOSTING REWARD

Here we provide a simple toy model in which we can prove rigorously that REGUIDANCE decreases the reconstruction loss (i.e., boosts the reward) achieved by the original sample $x$.

**Setup.** We consider the following mixture model with *exponentially many* modes. For a parameter $R > 0$, consider the uniform mixture of identity-covariance Gaussians centered at the points $\{R, -R\}^d$. Next, consider an inpainting-style linear measurement $A = (e_{i_1} \mid \cdots \mid e_{i_m})^\top$, where $e_i$ denotes the $i$-th standard basis vector in $\mathbb{R}^d$. Suppose we observed measurements $y \in \{R, -R\}^m$ which are consistent with some mode $x' \in \{R, -R\}^d$ (in fact $2^{d-m}$ many such modes), i.e. such that $y = Ax'$. Let $\Lambda$ be the affine subspace spanned by points consistent with this measurement.

Our first result shows that for small values of the hyperparameter $\sigma$, running our algorithm starting at a sample $x$ approximately in a sample $x_T^{\mathsf{DPS}}$ which is given by the projection of $x$ onto $\Lambda$. In other words, the reconstruction loss is driven to near zero under our algorithm:

**Theorem 1** (Informally *reward boosting*, see Theorem 4)**.** *If $q$ is given by a mixture of identity-covariance Gaussians centered at all $2^d$ points in $\{R, -R\}^d$, and $A \in \{0, 1\}^{m \times d}$ is an inpainting measurement, then on input $x \in \mathbb{R}^d$, REGUIDANCE outputs $x_T^{\mathsf{DPS}}$ for which $\|\Pi x - x_T^{\mathsf{DPS}}\| \leq \mathrm{poly}(\sigma, e^{-T})$, where $\Pi$ is the projection to the affine subspace given by all $x'$ for which $Ax' = y$.*

The phenomenon described in Theorem 1 is depicted in the left figure in Figure 2. Below, we briefly sketch the key idea, deferring the proof to Appendix C.1.

In the setting above, the DPS-ODE drift can be calculated explicitly. The denoiser can be written as

$$\mu_t(x) = e^t x_t + (e^t - e^{-t}) \nabla \ln q_t(x) = e^{-t} x + (1 - e^{-2t}) R \tanh(Re^{-t}x),$$

where $\tanh(\cdot)$ is applied entrywise, so

$$v_t^{\mathsf{DPS}}(x) = \left[ e^{-t} \mathrm{Id} + \frac{1 - e^{-2t}}{e^t} R^2 \mathrm{diag}(\mathrm{sech}^2(Re^{-t}x)) \right] \cdot \frac{A^\top (y - A\mu_t(x))}{\sigma^2}.$$

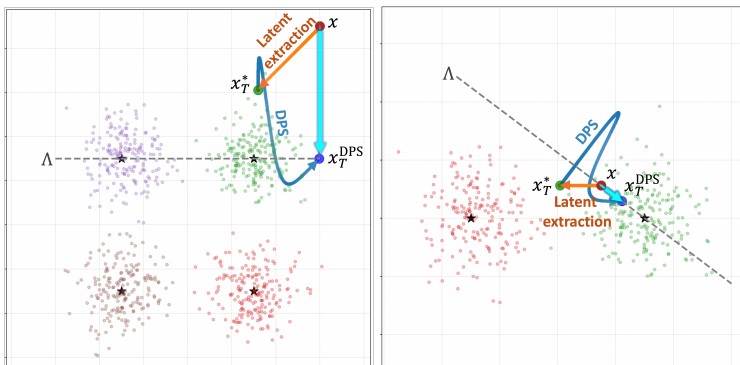

**Figure 2:** Initial reconstruction $x$ gets mapped via Step 1 of REGUIDANCE to latent $x_T^*$, then via Step 2 (DPS-ODE) to $x_T^{\text{DPS}}$. Left figure shows $x$ gets projected to subspace $\Lambda$ of maximal reward. Right figure shows even if $x$ is already on this subspace, the algorithm brings it closer to a mode, increasing likelihood / realism.

On the other hand, the other term in the velocity field for the DPS-ODE is $x + \nabla \ln q_t(x) = Re^{-t}\tanh(Re^{-t}x)$. In particular, as $\sigma \to 0$, the velocity field is dominated by $v_t^{\text{DPS}}$. And as $t$ approaches zero as the trajectory nears the end of the reverse process, the velocity field tends to

$$\lim_{t \to 0} v_t^{\text{DPS}}(x) = \frac{1}{\sigma^2} A^\top (y - Ax).$$

Note that if one projects $v_t^{\text{DPS}}$ to any of the directions $e_{i_j}$ among the rows of $A$, the projection is simply the score function of the one-dimensional Gaussian with mean $y_{i_j}$ and variance $\sigma^2$. Furthermore, one can verify that the dynamics of this ODE decouple across the coordinates, so that in each of the coordinates $i_j$ corresponding to a row of $A$, the corresponding one-dimensional ODE can be shown to converge to a small neighborhood around $y_{i_j}$. On the other hand, for all other coordinates $\ell \notin \{i_1, \ldots, i_m\}$, the corresponding one-dimensional ODE is the projection of the *unconditional* probability flow ODE to that coordinate, so by design, the dynamics converge to $x_\ell$.

We also show that one cannot replace the DPS-ODE in the second step of REGUIDANCE with the analogous *SDE* and achieve the same result, see Appendix C.2 for proof:

**Theorem 2** (Informal, see Theorem 5). *Let $q, A$ be as in Theorem 1. If the DPS-ODE in* REGUID-ANCE *is replaced by the analogous SDE, then even if $x$ is exactly equal to one of the modes in $\{R, -R\}^d$, running* REGUIDANCE *on $x$ results in $x_T^{\text{DPS}}$ which is bounded away from any of the modes in $\{R, -R\}^d$ with high probability over the randomness of the SDE.*

### 3.2 THEORY VIGNETTE 2: BOOSTING REALISM

While the above shows how our algorithm can boost reward, it says nothing about the extent to which it can bring $x$ towards regions of higher likelihood (realism). Indeed, as we will see in experiments on images (Section 4), our algorithm has the key benefit that even if the initial $x$ achieves reasonable reward, i.e. negligible reconstruction loss, it is still able to move it towards a more realistic $x_T^{\text{DPS}}$.

We now provide a simple toy model in which we can prove rigorously that REGUIDANCE boosts the likelihood of the sample under the data distribution. This is the most technically involved result in our paper, involving a multi-stage analysis of the DPS ODE dynamics, see Appendix D for details.

**Setup.** We consider a simple bimodal distribution. For a parameter $R > 0$, consider the uniform mixture of two identity-covariance Gaussians centered at the points $z_1 = Re_1$ and $z_2 = -Re_1$, where $e_1$ is the first standard basis vector (by rotation and translation invariance of our arguments below, the specific choice of means is without loss of generality). Consider a *single* linear measurement $A = v^\top \in \mathbb{R}^d$. Suppose we observed measurement $y = \langle v, z_1 \rangle$ consistent with mode $z_1$. Let $\Lambda$ be the affine hyperplane spanned by points consistent with this measurement.

We show that even if REGUIDANCE is applied to a point $x$ already on $\Lambda$, i.e. which already achieves maximal reward, under mild conditions our algorithm will result in a sample $x_T^{\text{DPS}}$ which is *closer* to the mode $z_1$ than $x$ is. In other words, the likelihood of the sample is boosted under our algorithm:

**Theorem 3** (Informally *realism boosting*, see Theorem 6). *If $q$ is given by a mixture of identity-covariance Gaussians centered at means $z_1 = Re_1$, $z_2 = -Re_1$, and $A = v^\top$ is an arbitrary single linear measurement, then on input $x$ satisfying $\langle v, x \rangle = y \triangleq \langle z_1, v \rangle$, provided that $\langle x, e_1 \rangle < R$ and $x$ is sufficiently close to $z_1$, there is an absolute constant $0 < C < 1$ such that the output $\overline{x}$ of a slight modification of* REGUIDANCE *satisfies $\|\overline{x} - z_1\| \leq C\|x - z_1\| + \mathrm{poly}(\sigma, e^{-T})$.*

**Remark 1.** Again, informally, for an initial sample $x$, Theorem 3 implies that the likelihood under the data distribution (realism) $p(x)$ strictly increases with REGUIDANCE. Theorem 1 boosts the reward, implying that the likelihood $p(y|x)$ of seeing the observed measurement $y$ also increases. By Bayes' Rule, it follows that REGUIDANCE increases the likelihood of $x$ conditioned on the measurement $y$; i.e., precisely increases the likelihood under the posterior $p(x|y)$.

The phenomenon described in Theorem 3 is depicted in the right figure in Figure 2.

## 4    Experiments on image data

**Datasets and models.** We primarily focus on the ImageNet $256 \times 256$ dataset[1] Deng et al. (2009) in the main body, and we defer additional experiments, including those conducted on the CIFAR-10 dataset Krizhevsky et al. (2009), to Appendix E. We use the pretrained unconditional $256 \times 256$ diffusion model from Dhariwal & Nichol (2021) as our base model.

**Inverse problems.** We focus on inverse problems with a well-defined measure of information loss induced by the measurement process. In particular, we consider the following tasks: *box-inpainting*, *super-resolution*, and *deblurring*. In box-inpainting, information loss is quantified by the size of the missing region; we consider two settings: *small inpainting*, where a random $128 \times 128$ region is removed from a $256 \times 256$ image, and *large inpainting*, where a random $191 \times 191$ region is removed. For super-resolution, information loss is quantified by the downsampling factor. *Small super-resolution* corresponds to $8\times$ resolution reduction, and *large super-resolution* to $16\times$ reduction. In these cases, we further corrupt the measurements by adding white Gaussian noise with standard deviation $0.05$. Finally, we examine *deblurring* tasks, where a blurring kernel is applied to the image Chung et al. (2023). In motion deblurring, the kernel is linear, while nonlinear deblurring uses a nonlinear corruption. The hard regime for these tasks corresponds to a high added Gaussian noise, which we set to have standard deviation $0.2$ (over the standard $0.05$).

**Baselines.** To show the effectiveness of REGUIDANCE, we use three posterior sampling methods: DDRM Kawar et al. (2022), DPS Chung et al. (2023), and DAPS Zhang et al. (2024). For each, we use a fixed evaluation set of 100 generated samples for each inverse task and apply REGUIDANCE. REGUIDANCE is implemented with the DDIM-based DPS Chung et al. (2023), setting the noise parameter $\eta$ to $0.0$. As a final (idealized) baseline, we apply REGUIDANCE on the *ground truth* images. Each run takes at most 7 GPU minutes on a single NVIDIA A100-SXM4-40GB GPU.

**Metrics.** We evaluate performance along the two axes of *measurement consistency* (reward) and *sample quality* (realism). For measurement consistency, we use the LPIPS Zhang et al. (2018) score of the generated sample relative to the ground truth. For sample quality, we use the well-known CMMD score Jayasumana et al. (2024), which exhibits very little variance on typical evaluation set sizes relative to other realism metrics Heusel et al. (2017), which can be unreliable in variance up to 5K+ samples. We use the standard 100 ImageNet samples for evaluation as used by DAPS Zhang et al. (2024) and Ye et al. (2024).

### 4.1    Inpainting

We present our main results for inpainting in Table 1. REGUIDANCE demonstrates large improvements in measurement consistency and sample quality across almost every baseline and task difficulty. As expected, REGUIDANCE informed with the ground truth latents has far superior LPIPS/-consistency scores, but surprisingly, DAPS and DDRM are able to yield latents that achieve superior realism scores. These strong empirical results are backed qualitatively in Figure 3, which shows that REGUIDANCE resolves high-level realism failures from the baselines and yields *diverse* and *high-quality* completions *distinct from the original image*. Notice that this type of behavior is *desired*.

---

[1]We choose ImageNet because unlike for other datasets in the literature (e.g., FFHQ), natural benchmarks for solving inverse problems with highly compressive measurements have yet to be saturated for this dataset.

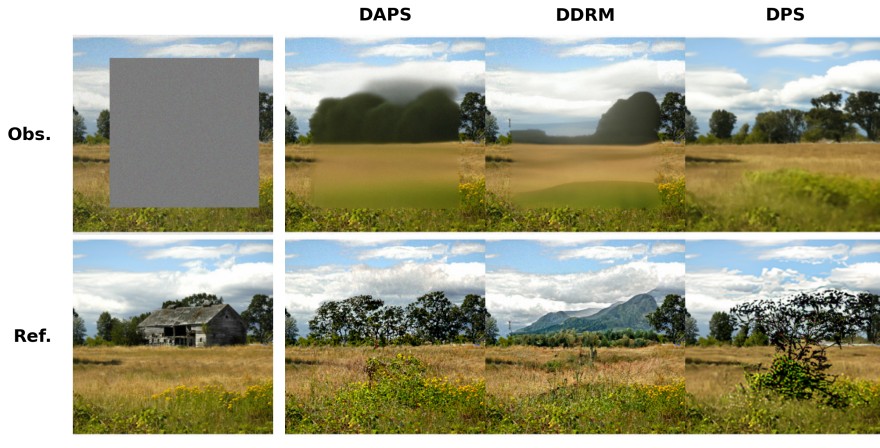

**Figure 3:** Examples of REGUIDANCE for inpainting with a $191 \times 191$ box. First column contains the observed measurement and reference image, while latter three demonstrate REGUIDANCE applied to different baselines.

Indeed, in highly compressive measurement regimes, recreating the original image (*data consistency*) is impossible, as multiple "ground truth" images can correspond to the same measurement. Instead, our method produces samples of high realistic quality that are consistent with the inpainted measurement, making them exemplary solutions to inverse inpainting. These dual objectives are precisely captured by the LPIPS and CMMD scores.

We provide additional visual examples in Appendix E.

| Method | Small Inpainting (Box) | | Large Inpainting (Box) | |
|---|---|---|---|---|
| | LPIPS ↓ | CMMD ↓ | LPIPS ↓ | CMMD ↓ |
| DAPS | 0.228 | 0.500 | 0.418 | 1.103 |
| DAPS + REGUIDANCE | **0.198** | **0.270** | **0.376** | **0.628** |
| DDRM | 0.238 | 0.664 | 0.426 | 1.164 |
| DDRM + REGUIDANCE | **0.199** | **0.263** | **0.372** | **0.616** |
| DPS | 0.270 | 0.525 | 0.393 | 0.720 |
| DPS + REGUIDANCE | **0.219** | **0.449** | **0.386** | 1.107 |
| Ground Truth + REGUIDANCE | 0.163 | 0.390 | 0.290 | 0.942 |

Table 1: Experimental results for box-inpainting. Bold numbers show improvements over baseline, and underlined values mark the best result (not including last row which uses knowledge of ground truth).

| Task | DAPS | | DAPS + REGUIDANCE | |
|---|---|---|---|---|
| | LPIPS ↓ | CMMD ↓ | LPIPS ↓ | CMMD ↓ |
| Superresolution (Small) | 0.410 | 1.257 | **0.404** | **1.186** |
| Superresolution (Large) | 0.545 | 2.114 | **0.496** | **1.582** |
| Motion Deblurring | 0.667 | 2.457 | **0.560** | **1.736** |
| Nonlinear Deblurring | 0.572 | 1.651 | **0.474** | **0.876** |

Table 2: Comparison of **DAPS** vs. **DAPS + REGUIDANCE** across various image restoration tasks. Best results for each metric/task pair highlighted in **bold**.

## 4.2 RESULTS ON OTHER IMAGE RESTORATION TASKS

To show the effectiveness of REGUIDANCE, we experiment with superresolution to $8\times$, superresolution to $16\times$, motion deblurring, and nonlinear deblurring. Given the broad improvements over

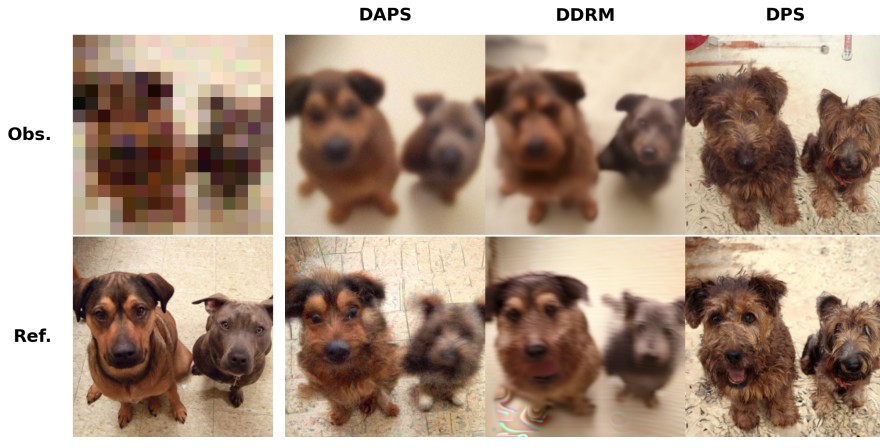

**Figure 4:** Examples of REGUIDANCE for $16\times$ super-resolution. First column contains the observed measurement and reference image, while latter three demonstrate REGUIDANCE applied to different baselines.

several baselines in the previous section, in this table, we restrict our initializations to the most state-of-the-art prior method, DAPS, and report our results in Table 2. We observe that REGUIDANCE consistently improves the performance for all the image restoration tasks. Additionally, we observe that the performance gap becomes larger for harder image restoration tasks (from 0.06 LPIPS score for $8\times$ superresolution to 0.49 for $16\times$ superresolution). Even in visual examples shown in Figure 4, we see that REGUIDANCE outputs images of much higher quality. Again, these images are necessarily similar yet not equivalent to the reference image; as they maintain high realistic quality and consistency with the blurred measurement, they are solutions to inverse super-resolution.

The qualitative results combined with visual examples confirm our hypothesis that REGUIDANCE improves both sample quality and measurement consistency when equipped with a strong initial latent. We provide additional visual examples for both superresolution and the deblurring tasks in Appendix E.

### 4.3 QUALITATIVE BEHAVIOR

In this section, we briefly describe observations on how REGUIDANCE benefits from the structure of good initial latents, the details of which we provide in Appendix E.3.

**SDE vs. ODE** The standard reverse SDE is known to be memoryless Domingo-Enrich et al. (2025), i.e., the generated sample does not depend on the initial latent. Adding Brownian motion to DPS similarly weakens the output's dependency on the latent, deteriorating performance as predicted by Theorem 2 and as we demonstrate in the supplement.

**Space of good latents.** We also demonstrate that the space of good initial latents is *disconnected*, allowing many possible sample initializations to provide *strong* and *diverse* quality boosts. While a small $L_2$ ball of latents around good latents also provide generally strong reconstructed samples, *it is not true* that all good latents are "neighbors" of each other, as we validate with images in the supplement.

## 5 OUTLOOK

In this work, we identify a difficult regime for steering diffusion models corresponding to highly compressive reward measurements. We introduce a simple and effective technique, REGUIDANCE, that leverages prior methods as strong latent initalizations for deterministic diffusion guidance, providing inverse problem solutions with boosted realism and reward. We also provide the first theoretical guarantees for DPS Chung et al. (2022b); it is an interesting future direction to extend these to richer families of data distributions. While we focused on reward guidance in the context of inverse problem solving, our technique is well-defined for other rewards. An immediate future direction is to evaluate the efficacy of REGUIDANCE for other choices of reward.

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

**Roadmap for appendix.**

- Appendix A: We discuss some other relevant works in the literature.

- Appendix B: For completeness, we prove the folklore result that DPS provably fails to sample from the correct tilted density even for very simple data distributions and measurements. To our knowledge, this has not yet appeared in written form in this literature.

- Appendix C: We give deferred proofs for our first and second main theoretical results. Theorem 1 shows in a stylized setting that REGUIDANCE projects the initial reconstruction to the manifold of maximal reward. Theorem 2 shows that if one replaces the DPS ODE in REGUIDANCE with the analogous SDE, it fails to achieve the desired property of boosting reward.

- Appendix D: We give deferred proofs for our third and most technically involved theoretical result, Theorem 3, which shows in a stylized setting that even if the initial reconstruction achieves maximal reward, REGUIDANCE moves it towards one of the modes of the distribution and thus increases its likelihood.

- Appendix E: We provide additional experimental results, including testing on alternate datasets, further results on superresolution, and visual examples of reconstructions generated using REGUIDANCE.

## A  FURTHER RELATED WORK

**Optimizing for latents.** The benefit of choosing the correct latent has been considered in prior works in the broader diffusion model literature Mokady et al. (2022); Huberman-Spiegelglas et al. (2024); Qi et al. (2024) for tasks like image editing, but it is not clear how these methods can be utilized for image restoration. To our knowledge, our work also gives the only known theoretical characterization of how the output of diffusion posterior sampling depends on the choice of latent.

**Theory for diffusion posterior sampling.** Posterior sampling with diffusion models is known to be computationally intractable in the worst case Gupta et al. (2024); Bruna & Han (2024). However, several recent works Xu & Chi (2024); Bruna & Han (2024); Montanari & Wu (2024); Karan et al. (2024) proposed algorithms with provable theoretical guarantees for posterior sampling under relaxed assumptions on the data distribution and/or measurements. These works are mainly for measurements that are either well-conditioned or have rank which is a constant fraction of the ambient dimension; in contrast, we work with highly compressive measurements. The exception is the algorithm of Xu & Chi (2024) which comes with rigorous guarantees for general inverse problems; their results however are asymptotic in nature, and they evaluated on superresolution on ImageNet $256 \times 256$ but only at $4\times$ downsampling, compared to $8\times, 16\times$ in our work.

## B  DPS FAILS TO SAMPLE FROM THE TILTED DENSITY

In this section we provide a proof of the folklore result that DPS provably fails to sample from the correct tilted density even for very simple data distributions and measurements, and even with perfect score estimation. We essentially prove that the KL divergence between the distribution of the DPS algorithm and the correct tilted density is large.

Consider running the reverse SDE corresponding to the correct SDE and SDE given by the DPS:

$$\mathrm{d}x_t = (x_t + 2\nabla \ln q_{T-t}(x_t) + 2\nabla \ln q_{T-t}(y|x_t))\mathrm{d}t + \sqrt{2}\mathrm{d}B_t \tag{1}$$

$$\mathrm{d}x_t^{\mathsf{DPS}} = (x_t^{\mathsf{DPS}} + 2\nabla \ln q_{T-t}(x_t^{\mathsf{DPS}}) + 2v_{T-t}^{\mathsf{DPS}}(x_t^{\mathsf{DPS}}))\mathrm{d}t + \sqrt{2}\mathrm{d}B_t. \tag{2}$$

We first show that the drift terms of eq.(1) and eq.(2) are different when $q(x) = \mathcal{N}(x; 0, \mathrm{Id})$ and $y = x + \sigma z$ where $z \sim \mathcal{N}(0, \mathrm{Id})$. Observe that $\nabla \ln q_{T-t}(x) = -x$ for all $t$. Using $x_t = e^{-t}x_0 + \sqrt{1 - e^{-2t}}z$ for $z \sim \mathcal{N}(0, \mathrm{Id})$, the joint distribution of $[x_0, x_t]$ is given by

$$p\begin{pmatrix} x_0 \\ x_t \end{pmatrix} = \mathcal{N}\left( \begin{pmatrix} 0 \\ 0 \end{pmatrix}, \begin{pmatrix} \mathrm{Id} & e^{-t}\mathrm{Id} \\ e^{-t}\mathrm{Id} & \mathrm{Id} \end{pmatrix} \right).$$

The conditional probability $p(x_0|x_t)$ is a Gaussian distribution with mean $e^{-t}x_t$ and covariance $(1-e^{-2t})\text{Id}$. Using $p(y|x_0) \propto \exp(-\frac{\|y-x_0\|^2}{2\sigma^2})$ and $p(x_0|x_t) \propto \exp(-\frac{\|x_0-e^{-t}x_t\|^2}{2(1-e^{-2t})})$, the probability density $p(y|x_t) = \int p(x_0|x_t)p(y|x_0)\mathrm{d}x_0 \sim \mathcal{N}(y; e^{-t}x_t, \sigma^2 + 1 - e^{-2t})$. Using this, the score function

$$\nabla \ln q_{T-t}(y|x_t) = e^{-t}(y - e^{-t}x_t)/(\sigma^2 + 1 - e^{-2t}).$$

When $q(x) = \mathcal{N}(x; 0, \text{Id})$, $\mathbb{E}[x_0|x_t] = e^{-t}x_t$. In this case, the DPS term in (2) is given by

$$v_{T-t}^{\mathsf{DPS}}(x) = \nabla \ln q_{T-t}(y \mid \mathbb{E}[x_0|x_t = x]) = e^{-t}(y - e^{-t}x)/\sigma^2.$$

This proves that the two drift terms of the correct conditional SDE (1) and DPS SDE (2) are different. Now, to prove that these two SDEs result in different distributions, we use Girsanov's theorem. We first check Novikov's condition required for Girsanov's theorem. First, observe that $\|v_{T-t}^{\mathsf{DPS}}(x) - \nabla \ln q_{T-t}(y|x_t)\| = (1 - e^{-2t})e^{-t}\|y - e^{-t}x_t\|/(\sigma^2(\sigma^2 + 1 - e^{-2t}))$. We also have $p(x_t|y) = \mathcal{N}(x_t; e^{-t}y/(\sigma^2 + 1), (\sigma^2 + 1 - e^{-2t})/(\sigma^2 + 1))$. By rewriting $x_s \sim p(x_s|y)$ in terms of standard Gaussian $z$, we obtain

$$\|v_{T-t}^{\mathsf{DPS}}(x) - \nabla \ln q_{T-t}(y|x_t)\| = \frac{(1 - e^{-2t})e^{-t}\|y - e^{-t}x_t\|}{\sigma^2(\sigma^2 + 1 - e^{-2t})}$$

$$\leq \frac{\|y\| + \|(\sigma^2 + 1 - e^{-2t})^{0.5}/(\sigma^2 + 1)^{0.5}z\|}{\sigma^2(\sigma^2 + 1 - e^{-2t})}$$

Using this upper bound, we can check that Novikov's condition holds when $T \leq \sigma^8/8$:

$$\mathbb{E}_{x_s \sim q_{T-s}(\cdot|y)}\Big[\exp\Big(2\int_0^T \|v_{T-s}^{\mathsf{DPS}}(x_s) - \nabla \ln q_{T-s}(y|x_s)\|^2 \mathrm{d}s\Big)\Big]$$

$$\leq \mathbb{E}_{z \sim \mathcal{N}(0, \text{Id})}\Big[\exp\Big(4T\Big(\frac{\|y\|^2}{\sigma^8} + \frac{\|z\|^2}{\sigma^8}\Big)\Big)\Big] < \infty$$

Let $Q_T^{\mathsf{DPS}}(x|y)$ and $Q_T(x|y)$ be the path measure of the DPS algorithm (2) and correct conditional SDE (1), respectively. Then, using Girsanov's theorem, for a fixed $y$, the KL divergence between

$$KL\big(Q_T(x|y)\|Q_T^{\mathsf{DPS}}(x|y)\big) = \mathbb{E}_{Q_T(x|y)}\int_0^T \|v_{T-t}^{\mathsf{DPS}}(x) - \nabla \ln q_{T-t}(y|x_t)\|^2 \mathrm{d}t$$

$$= \mathbb{E}_{Q_T(x|y)}\Big[\int_0^T \frac{(1 - e^{-2t})^2 e^{-2t}\|y - e^{-t}x_t\|^2}{\sigma^4(\sigma^2 + 1 - e^{-2t})^2}\mathrm{d}t\Big]$$

$$= \int_0^T \frac{(1 - e^{-2t})^2 e^{-2t}}{\sigma^4(\sigma^2 + 1 - e^{-2t})^2}\Big(\big(1 - \frac{e^{-2t}}{\sigma^2 + 1}\big)\|y\|^2 + d\frac{e^{-2t}(\sigma^2 + 1 - e^{-2t})}{\sigma^2 + 1}\Big)\mathrm{d}t$$

By changing variable using $x = e^{-2t}$, we obtain

$$KL\big(Q_T(x|y)\|Q_T^{\mathsf{DPS}}(x|y)\big) \geq \int_{e^{-2T}}^1 \frac{(1-x)^2}{2\sigma^4(\sigma^2 + 1)^2}(\|y\|^2 + dx)\mathrm{d}t \geq \frac{\|y\|^2(1 - e^{-2T})^3}{6\sigma^4(\sigma^2 + 1)^2}$$

## C   DEFERRED PROOFS FROM SECTION 3.1

### C.1   PROOF OF THEOREM 1

We state and prove the following formal version of Theorem 1, which states that when initialized at a point which does not achieve maximal reward, REGUIDANCE will project that point to the subspace of points that achieve maximal reward.

**Theorem 4.** *Let $q$ be the uniform mixture of identity-covariance Gaussians centered at all $2^d$ points in $\{R, -R\}^d$, and let $A \in \{0,1\}^{m \times d}$ be an inpainting measurment, i.e. a Boolean matrix with exactly one nonzero entry in each row. Suppose we observe the measurement $y = Ax^*$, where $x^* \in \{R, -R\}^d$. Let $x_T^{\mathsf{DPS}}$ denote the output of REGUIDANCE with guidance strength $\rho = 1/\sigma^2$ and time $T$ starting from initial reconstruction $x$. Then $\|\Pi x - x_T^{\mathsf{DPS}}\| \leq poly(\sigma Rd\|y\|, TRd\|y\|e^{-T})$, where $\Pi$ is the projection to the affine subspace of all $x'$ for which $Ax' = y$.*

*Proof of Theorem 4.* Recall that starting from an initial construction $x$, the REGUIDANCE algorithm runs the following ODE for $T$ duration:

$$\mathrm{d}x_t^{\mathsf{DPS}} = \left(x_t^{\mathsf{DPS}} + \nabla \ln q_{T-t}(x_t^{\mathsf{DPS}}) + \rho \nabla \mu_{T-t}(x_t^{\mathsf{DPS}}) A^\top (y - A\mu_{T-t}(x_t^{\mathsf{DPS}}))\right) \mathrm{d}t, \quad x_0^{\mathsf{DPS}} = x.$$

Consider the uniform mixture of identity-covariance Gaussians centers at the points $\{R, -R\}^d$ for a parameter $R > 0$. For this distribution, the score function at noise scale $t$ is given by

$$\nabla \log q_t(x) = \sum_{j=1}^{d} (-x_j + Re^{-t} \tanh(Re^{-t}x_j))e_j$$

where $\{e_1, \ldots, e_d\}$ are standard basis vector in $\mathbb{R}^d$. In other words, in every direction, the score function is $-x + Re^{-t} \tanh(Re^{-t}x)$. The drift of the ODE can be decomposed into two parts: unconditional drift, denoted by $U_{T-t}(x_t)$, and conditional DPS term, denoted by $v_{T-t}^{\mathsf{DPS}}(x_t)$. The unconditional and conditional term for $i^{th}$ coordinate is given by

$$U_{T-t}(x)_i = -x_i + Re^{-(T-t)} \tanh(Re^{-(T-t)}x_i)$$

$$v_{T-t}^{\mathsf{DPS}}(x)_i = \left[\left(e^{-(T-t)}\mathrm{Id} + (1 - e^{-2(T-t)})e^{-(T-t)}R^2 \mathrm{diag}(\mathrm{sech}^2(Re^{-(T-t)}x))\right) \cdot \frac{A^\top(y - A\mu_{T-t}(x))}{\sigma^2}\right]_i$$

$$= \begin{cases} e^{-(T-t)}(1 + (1 - e^{-2(T-t)})R^2\mathrm{sech}^2(Re^{-(T-t)}x))\frac{(y - \mu_{T-t}(x))}{\sigma^2} & \text{if } e_i \text{ is in subspace of } A \\ 0 & \text{otherwise} \end{cases}$$

We refer to coordinates that are observed/measured as 'measured coordinates' and 'unmeasured coordinates' if they are not observed. We first focus on the analysis of 'measured coordinates'. Let $i$ be such a coordinate and $y_i$ be the measurement of it. For brevity, we will drop the superscript DPS and the coordinate index $i$ from the subscript as the analysis applies to any measured coordinate. For these coordinates, we track the evolution of error $\mathrm{d}(y - x_t)^2/\mathrm{d}t$ as follows:

$$\frac{\mathrm{d}(y - x_t)^2}{\mathrm{d}t} = -2(y - x_t)\frac{\mathrm{d}x_t}{\mathrm{d}t} \tag{3}$$

We first rewrite $(y - \mu_{T-t}(x))$ as follows:

$$y - \mu_{T-t}(x) = (y - x) + (x - e^{-(T-t)}x) - (1 - e^{-2(T-t)})R\tanh(Re^{-(T-t)}x)$$

Using this, we can rewrite $v_{T-t}^{\mathsf{DPS}}(x_t) = e^{-(T-t)}\left(\frac{1}{\sigma^2} + \frac{(1 - e^{-2(T-t)})R^2\mathrm{sech}^2(Re^{-(T-t)}x)}{\sigma^2}\right)(y - x_t) + E_t$ where the additional term $E_t$ is given by

$$E_t = e^{-(T-t)}\frac{\left((x - e^{-(T-t)}x) - (1 - e^{-2(T-t)})R\tanh(Re^{-(T-t)}x)\right)}{\sigma^2}$$

$$+ e^{-(T-t)}(1 - e^{-2(T-t)})R^2\mathrm{sech}^2(Re^{-(T-t)}x)\frac{(x - \mu_{T-t}(x))}{\sigma^2}$$

Combining this with (3), we obtain

$$\frac{\mathrm{d}(y - x_t)^2}{\mathrm{d}t} = -2e^{-(T-t)}\left(\frac{1}{\sigma^2} + \frac{(1 - e^{-2(T-t)})R^2\mathrm{sech}^2(Re^{-(T-t)}x)}{\sigma^2}\right)(y - x_t)^2 - 2(y - x_t)(U_t + E_t)$$

It is easy to prove that $|U_t| \leq R$ and the upper bound on $|E_t|$ is given by

$$|E_t| \leq \frac{2R^2 e^{-(T-t)}}{\sigma^2}(1 - e^{-(T-t)})|x_t| + \frac{2R^3 e^{-(T-t)}}{\sigma^2}(1 - e^{-2(T-t)}) + e^{-(T-t)}(1 - e^{-2(T-t)})|y - x_t|$$

Let $r_t$ be equal to $|y - x_t|$. Then, using $|x_t| \leq r_t + |y|$ and $-\mathrm{sech}^2(Re^{-(T-t)}x) \leq 0$, for $t \in [T - \sigma, T]$ we obtain

$$\frac{\mathrm{d}r_t}{\mathrm{d}t} \leq a(t)r_t + b(t)$$

$$\text{where } a(t) = -\frac{e^{-(T-t)}}{\sigma^2} + (1 - e^{-2(T-t)}) \leq -\frac{e^{-\sigma}}{\sigma^2} + 2\sigma,$$

$$b(t) = R + \frac{2R^3}{\sigma^2}(1 - e^{-2(T-t)}) + \frac{2R^2}{\sigma^2}(1 - e^{-(T-t)})|y|$$

$$\leq R + \frac{4R^3}{\sigma} + \frac{2R^2}{\sigma}|y|,$$

where the above inequality follows from $(1 - e^{-z}) \leq z$ for all $z$. We now integrate the above inequality from $t = T - \sigma$ to $T$ for sufficiently small $\sigma$ (i.e., $\sigma \leq 0.1$). For such $\sigma$, $a(t) \leq -1/10\sigma^2$. For $R \geq 1$, this gives us that

$$r_T \lesssim e^{-\frac{1}{10\sigma}}(r_{T-\sigma} + \sigma\max(R^3, R^2|y|)(e^{\frac{1}{10\sigma}} - 1))$$

$$\lesssim e^{-\frac{1}{10\sigma}}r_{T-\sigma} + \sigma\max(R^3, R^2|y|)$$

We now prove an upper bound on $r_{T-\sigma}$ by proving an upper bound on $|x_{T-\sigma}|$. Using the conditional DPS and unconditional term, for each coordinate, we have

$$\frac{\mathrm{d}x_t}{\mathrm{d}t} = Re^{-(T-t)}\tanh(Re^{-(T-t)}x_i) + e^{-(T-t)}(1 + (1 - e^{-2(T-t)})R^2\mathrm{sech}^2(Re^{-(T-t)}x_t))\frac{(y - \mu_{T-t}(x_t))}{\sigma^2}$$

$$\leq R + \frac{R^2}{\sigma^2}(|y| - e^{-(T-t)}x_t) + \frac{R^3}{\sigma^2}.$$

Solving this inequality for $t = 0$ to $t = T - \sigma$, we have

$$x_{T-\sigma} \leq e^{-\frac{R^2}{\sigma^2}(e^{-\sigma} - e^{-T})}\left[x_0 + \left(\frac{R^2|y| + R^3}{\sigma^2}\right)\int_0^{T-\sigma} e^{\frac{R^2}{\sigma^2}(e^{-(T-s)} - e^{-T})}\mathrm{d}s\right] \leq x_0 + \frac{(R^2|y| + R^3)T}{\sigma^2}.$$

Using this bound, we obtain

$$r_T \lesssim e^{-\frac{1}{10\sigma}}\left(x_0 + \frac{(R^2|y| + R^3)T}{\sigma^2}\right) + \sigma\max(R^3, R^2|y|).$$

For sufficiently small $\sigma \leq 1/T$, we obtain that for every measured coordinate, we have $|y - x_T| \leq \mathrm{poly}(\sigma R|y|, TR|y|e^{-T})$. For every unmeasured coordinate, the velocity field remains the same as the unconditional score function. Therefore, the unmeasured coordinates converge to the same value as the initial reconstruction after REGUIDANCE. Combining these two claims, we obtain the result. $\qquad\square$

## C.2 PROOF OF THEOREM 2

In this section, we state and prove the following formal version of Theorem 2, which states that the guarantees of REGUIDANCE from Theorem 1 do not hold if one replaces the ODE with an SDE:

**Theorem 5.** *Let $q$ be the uniform mixture of identity-covariance Gaussians centered at all $2^d$ points in $\{R, -R\}^d$, and let $A \in \{0,1\}^{m \times d}$ be an inpainting measurement, i.e. a Boolean matrix with exactly one nonzero entry in each row. Let $y = Ax$ be the value of the measurement for some $x \in \{R, -R\}^d$. If REGUIDANCE is run on the (perfect) initial reconstruction $x$ but using the SDE instead of the ODE, then the result $x_T^{\mathsf{DPS}}$ is independently distributed as $\frac{1}{2}\mathcal{N}(R, 1) + \frac{1}{2}\mathcal{N}(-R, 1)$, up to $\exp(-\Omega(T))$ statistical error, on all other coordinates.*

*Proof.* Recall from the previous section that the DPS term $v_t^{\mathsf{DPS}}$ is zero on the unmeasured coordinates, i.e. the coordinates corresponding to zero columns of $A$. Furthermore, as observed in the previous section, the dynamics of the DPS-SDE decouple across coordinates. In the unmeasured coordinates, the velocity field is simply given by the *unconditional score* for the mixture $\frac{1}{2}\mathcal{N}(R, 1) + \frac{1}{2}\mathcal{N}(-R, 1)$ marginalized to that coordinate. That is, it is given by the SDE

$$\mathrm{d}x_t = (-x_t + 2Re^{-(T-t)}\tanh(Re^{-(T-t)}x_t))\,\mathrm{d}t + \sqrt{2}\mathrm{d}B_t.$$

This SDE is initialized at the $i$-th coordinate of the vector $x$, which we denote by $x[i]$, and the final iterate $x_T$ is a sample from the posterior distribution over $x \sim \frac{1}{2}\mathcal{N}(R,1) + \frac{1}{2}\mathcal{N}(-R,1)$ conditioned on $e^{-T}x + \sqrt{1 - e^{-2T}}g = x[i]$ for $g \sim \mathcal{N}(0, \mathrm{Id})$. The KL divergence between this posterior and $\frac{1}{2}\mathcal{N}(R,1) + \frac{1}{2}\mathcal{N}(-R,1)$ decays exponentially with $T$, as claimed. $\qquad\square$

# D    PROOF OF THEOREM 3

## D.1    PROOF PRELIMINARIES

We begin by giving the formal version of Theorem 3, which states that when initialized at a point with maximal reward, under some mild conditions REGUIDANCE will contract that point even closer to one of the modes of the data distribution. To do this, we will first introduce a modification of REGUIDANCE to assist with the formalization. Throughout, given a vector $w$, we will use the shorthand $w[i]$ to denote $\langle w, e_i \rangle$, where $e_i$ is the $i$-th standard basis vector.

**Modification of REGUIDANCE.**    Here we explicitly compute the velocity field of the ODE that we run. Let $q_t$ denote the marginal at time $t$ of running the Ornstein-Uhlenbeck process starting at $q$. Note that

$$x + \nabla \ln q_t(x) = Re^{-t} \tanh(Re^{-t}x[1]) \cdot e_1$$

The denoiser $\hat{x}_0(x_t)$ is given by

$$e^{-t}x_t + (1 - e^{-2t})R \tanh(Re^{-t}x_t[1]) \cdot e_1 \,,$$

and the Jacobian of the denoiser is given by

$$e^{-t}\mathrm{Id} + \frac{1 - e^{-2t}}{e^t}R^2 \mathrm{diag}(\{\mathrm{sech}^2(Re^{-t}x[1]), 0, \ldots, 0\})\,.$$

The DPS term in the velocity field of the DPS-ODE is thus given by the product

$$\frac{1}{\sigma^2}\left[e^{-t}\mathrm{Id} + \frac{1 - e^{-2t}}{e^t}R^2 \mathrm{diag}(\{\mathrm{sech}^2(Re^{-t}x[1]), 0, \ldots, 0\})\right]\cdot$$
$$\left(vv^\top (Re_1 - e^{-t}x_t - (1 - e^{-2t})R \tanh(Re^{-t}x_t[1]) \cdot e_1)\right)\,.$$

The $\mathrm{sech}^2$ term is unnecessarily cumbersome for our analysis and does not qualitatively impact the behavior of REGUIDANCE in this setting, so we define MODIFIEDREGUIDANCE to be given by the following ODE:

$$\mathrm{d}x_t^{\mathsf{MDPS}} = \Big\{ Re^{-(T-t)} \tanh(Rx_t^{\mathsf{MDPS}}[1])e_1$$
$$+ \frac{e^{-(T-t)}}{\sigma^2}vv^\top \big(Re_1 - x_t^{\mathsf{MDPS}}$$
$$- (1 - e^{-2(T-t)})R \tanh(Rx_t^{\mathsf{MDPS}}[1])e_1\big)\Big\}\, \mathrm{d}t\,. \qquad \text{(MDPS-ODE)}$$

We are now ready to state the formal version of Theorem 3.

**Theorem 6.** *Let $q$ be a mixture of two Gaussians $q = \frac{1}{2}\mathcal{N}(z_1, \mathrm{Id}) + \frac{1}{2}\mathcal{N}(z_2, \mathrm{Id})$ where $z_1 = Re_1$ and $z_2 = -Re_1$. For measurement $y = \langle z_1, v \rangle$ given by a single unit vector $A = v^\top$, let $x_T^{\mathsf{MDPS}}$ denote the output of MODIFIEDREGUIDANCE (Eq. (MDPS-ODE)) with guidance strength $\rho = 1/\sigma^2$ and time $T$ starting from initial reconstruction $x$ which has maximal reward, i.e., which satisfies $y = \langle x, v \rangle$.*

*There is an absolute constant $c > 0$ such that the following holds. Suppose $\langle x_0^{\mathsf{MDPS}}, v \rangle \le cR\langle v, e_1 \rangle$, where $x_0^{\mathsf{MDPS}} = x_T^*$ is the latent noise vector given by running the unconditional probability flow ODE in reverse starting from $x$. Furthermore, suppose that $\langle x, e_1 \rangle < R$; $\sigma \ll 1$; $T \gg \log(R/\sigma)$; and $\langle v, e_1 \rangle$ is bounded away from $\{0, 1, -1\}$. Then:*

- ***Reward approximately preserved**: $\frac{1}{R}|\langle x_T^{\mathsf{MDPS}}, v \rangle - y| \lesssim \sigma \log(1/\sigma)\langle v, e_1 \rangle$*

- ***Contraction toward mode**: $\|x_T^{\mathsf{MDPS}} - z_1\| \le C\|x - z_1\|$ for a factor $0 < C < 1$ which tends towards $\langle v, e_1 \rangle^2$ as $\sigma \to 0$.*

Before proceeding to the proof, we provide some simplifying reductions.

**Reparametrization.** Instead of tracking $x_t^{\mathsf{MDPS}}$, we will track $x_t' \triangleq e^{-(T-t)}x_t^{\mathsf{MDPS}}$. Under this change of variable, Eq. (MDPS-ODE) becomes

$$\mathrm{d}x_t' = \Big\{ x_t' + Re^{-2(T-t)}\tanh(Rx_t'[1])e_1$$

$$+ \frac{e^{-2(T-t)}}{\sigma^2}vv^\top\big(Re_1 - x_t' - (1 - e^{-2(T-t)})R\tanh(Rx_t'[1])e_1\big)\Big\}\,\mathrm{d}t\,.$$

**Reduction to two-dimensional problem.** Note that the drift only depends on the projection of the trajectory to the two-dimensional subspace spanned by $v$ and $e_1$. Furthermore, the construction of the latent noise vector $x_T^*$ out of initial reconstruction $x$ is given by running an ODE which decouples across every coordinate vector $e_i$. We will thus henceforth assume without loss of generality that the original Gaussian mixture was two-dimensional by projecting to the span of $v$ and $e_1$. Extend $e_1$ to an orthonormal basis for this subspace; we will write occasionally express points in the coordinates given by this basis, e.g. $v = (v[1], v[2])$.

We can always assume without loss of generality that $v[1] \geq 0$. Furthermore, as the data distribution is symmetric under reflection around $e_1$, we can additionally assume without loss of generality that $v[2] \geq 0$.

Additionally, let $v^\perp = (-v[2], v[1])$ denote the orthogonal complement of $v$.

**Latent noise vector** Let $x_T^*$ denote the latent noise vector obtained by starting from initial reconstruction $x$ and running the unconditional probability flow ODE in reverse for time $T$.

**Lemma 1.** $x_T^* = (c, x[2])$ *for some $c$ for which* $\mathrm{sgn}(c) = \mathrm{sgn}(x[1])$.

We defer the proof to Appendix D.

### D.2 Proof Overview

Our high-level strategy will be to argue that for most of the trajectory, $\tanh(Rx_t'[1])$ is very close to 1. Note that the drift of the process $(x_t^{\mathsf{MDPS}})$ in the direction $v^\perp$ is $Re^{-2(T-t)}\tanh(Re^{-(T-t)}x_t^{\mathsf{MDPS}}[1])$ and thus entirely dictated by $(x_t^{\mathsf{MDPS}}[1])$. Provided that the latter is close to 1 for most of the trajectory, this ensures that the total movement in the $v^\perp$ direction is close to $-Rv[2]$ (see Eq. (8) below). We will also need to track the total movement in the $v$ direction; in fact in the course of analyzing the movement in this direction, we will show that $\tanh(Rx_t'[1]) \approx 1$ for most of the trajectory as a byproduct.

Roughly speaking, our analysis of the trajectory of MODIFIEDREGUIDANCE can be broken into three stages, depicted in Figure 5:

- **Stage 1**: In this stage, $\tanh(Rx_t'[1])$ increases from a small value to nearly 1 over a time window of length roughly $T - O(\log(R/\sigma))$ (see the definition of $T_1'$ below). At the same time, as $x_t'[1]$ is increasing, the quantity $\langle x_t', v\rangle$ is increasing but not too quickly, up to a value of at most $\tilde{O}(\sigma\sqrt{R})$.
- **Stage 2**: In this stage, from time $T - O(\log(Rv[1]/\sigma))$ to time $T - \log(1/\sigma)$, $x_t'[1]$ is still increasing, and $\langle x_t', v\rangle$ steadily increases to a value of at most $\tilde{O}(\sigma R)$.
- **Stage 3**: In the final stage, from time $T - \log(1/\sigma)$ to time $T$, $\tanh(Rx_t'[1])$ is no longer monotonically increasing but also never drops below $1 - O(\sigma)$. Because the value of $\tanh(Rx_t'[1])$ is still close to 1, the evolution of $\langle x_t', v\rangle$ is well-approximated by a self-consistent evolution purely in the direction of $v$ (see Eq. (12)). The form of this evolution can be then be used both to derive a good estimate for the final value $\langle x_T', v\rangle$, and also to control the fluctuations of $\tanh(Rx_t'[1])$ around 1 in this final stage.

Altogether, this three-stage analysis will allow us to prove the following key lemma:

**Lemma 2** (Dynamics of REGUIDANCE). *Let*

$$\delta' \triangleq e^{-2(T-T_1')} \asymp \frac{\log(1/\epsilon)\sigma^2}{Rv[1]}\,. \tag{6}$$

*If $x_0^{\mathsf{MDPS}}[1] \geq 0$, then:*

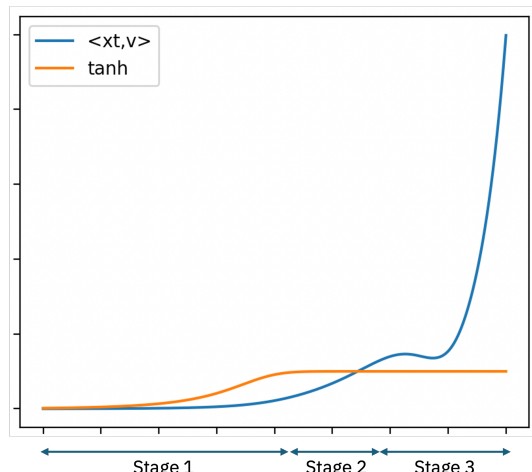

**Figure 5:** Depiction of evolution of $\tanh(Rx'_t)$ and $\langle x'_t, v \rangle$ and the three stages of analysis

1. $x_t^{\mathsf{MDPS}}[1] \geq 0$ *for all* $0 \leq t \leq T$.

2. *For* $t \geq T - \log(1/\delta')$, $\tanh(Re^{-(T-t)} x_t^{\mathsf{MDPS}}[1]) \geq 1 - O(\sigma)$.

3. $|\langle x_T^{\mathsf{MDPS}}, v \rangle - y| \lesssim Rv[1]\sigma \log(1/\sigma)$.

The proof of this is quite subtle and is the crux of our argument. We defer the proof to Appendix D.4. With Lemma 2 in hand, we can now conclude the proof of our main result.

*Proof of Theorem 6.* The guarantee on reward follows from Part 3 of Lemma 2.

We now establish contraction towards the mode. Recall from Section D.1 the definition of $v^\perp = (-v[2], v[1])$ and the fact that we are assuming without loss of generality that $v[1], v[2] \geq 0$.

Let us compute the total movement in the direction of $v^\perp$ over the course of the DPS ODE trajectory. For this calculation, let us work with the original process $(x_t^{\mathsf{MDPS}})$ rather than the reparametrized one. The drift in the $v^\perp$ direction is given by

$$\left\langle \frac{\mathrm{d}x_t^{\mathsf{MDPS}}}{\mathrm{d}t}, v^\perp \right\rangle = -Re^{-(T-t)} \tanh(Re^{-(T-t)} x_t^{\mathsf{MDPS}}[1]) \cdot v[2]. \tag{7}$$

In the first two parts of Lemma 2, it was shown that if $x_0^{\mathsf{MDPS}}[1] \geq 0$, then $x_t^{\mathsf{MDPS}}[1] \geq 0$ for all $0 \leq t \leq T$. Furthermore, for $t \geq T'_1$, $\tanh(Re^{-(T-t)} x_t^{\mathsf{MDPS}}[1]) \geq 1 - O(\sigma)$.

Integrating Eq. (7), we conclude that

$$\langle x_T^{\mathsf{MDPS}}, v^\perp \rangle - \langle x_0^{\mathsf{MDPS}}, v^\perp \rangle \leq (1 - O(\sigma))v[2] \cdot \int_{T-\log(1/\delta')}^{T} -Re^{-(T-t)} \, \mathrm{d}t$$

$$= (1 - O(\sigma))Rv[2] \cdot (1 - \delta')$$

$$\leq -(1 - O(\sigma))Rv[2]. \tag{8}$$

Recall that $x_0^{\mathsf{MDPS}} = x_T^* = (c, x[2])$ for $c$ defined in Lemma 1 and $x[2]$ the second coordinate of the initial reconstruction $x$. At the end of the trajectory, $\langle x_T^{\mathsf{MDPS}}, v^\perp \rangle = -(c + R)v[2] + v[1]x[2] \pm O(R\sigma v[2])$ and $\langle x_T^{\mathsf{MDPS}}, v \rangle = (1 \pm \sigma \log(1/\sigma))Rv[1]$.

Because $x$ was assumed to achieve maximal reward $\langle v, x \rangle = Rv[1]$, so that $x[1] = R - \frac{v[2]x[2]}{v[1]}$ (here we are using that $v[1]$ is bounded away from zero) and thus

$$\langle x, v^\perp \rangle = -v[2]x[1] + v[1]x[2] = -Rv[2] + \frac{v[2]^2x[2]}{v[1]} + v[1]x[2].$$

Note that $x[2] > 0$ because we are assuming in Theorem 6 that $x[1] < R$ and furthermore we are assuming $v[1], v[2] \geq 0$ without loss of generality.

In contrast, we have

$$\langle z_1, v^\perp \rangle = -Rv[2]$$

We thus have

$$\|z_1 - x\| = |\langle x - z_1, v^\perp \rangle| \leq \frac{v[2]^2 x[2]}{v[1]} + v[1]x[2].$$

On the other hand,

$$\|z_1 - x_T^{\mathsf{MDPS}}\| \leq \sqrt{(-cv[2] + v[1]x[2] + O(R\sigma v[2]))^2 + R^2 v[1]^2 \sigma^2 \log^2(1/\sigma)}$$
$$\leq O(Rv[1]\sigma \log(1/\sigma)) + v[1]x[2].$$

Provided that $x[2] \gtrsim R\sigma \log(1/\sigma)v[1]^2/v[2]^2$, we get a contraction from $\|z_1 - x\|$ to $\|z_1 - x_T^{\mathsf{MDPS}}\|$. Note that in particular, as $\sigma \to 0$, the factor of contraction tends towards $v[1]^2$ as claimed. $\square$

### D.3 Proof of Lemma 1

The unconditional probability flow ODE in reverse time is given by

$$\mathrm{d}x_t^* = -Re^{-t} \tanh(Re^{-t}x_t^*[1]) \, \mathrm{d}t$$

First, note that the velocity field does not depend on $x_t^*[2]$, so $x_T^*[2] = x[2]$. Next, note that the unconditional probability flow ODE in *forward time* is monotone in the sense that it ends up at a point whose first coordinate is the same sign as that of its initialization. The fact that $c$ in the claim has the same sign as $x[1]$ follows by reversing time.

### D.4 Proof of Lemma 2

In preparation for the analysis, define

$$\epsilon = 4\sigma^2 \qquad (9)$$

and define the stopping times

- $T^*$: first time that $\langle x_t', v \rangle > \epsilon Rv[1]/2$.
- $T_1 = T - \frac{1}{2}\log(1/\delta)$ for $\delta \triangleq \frac{3\sigma^2}{Rv[1]}$
- $T_1' = T_1 + \Theta(\log\log 1/\epsilon)$
- $T_2 = T - \log(1/\sigma)$

#### D.4.1 Stage 1: $t = 0$ to $t = T_1' = T - O(\log(R/\sigma))$

While $\tanh(Rx_t'[1]) \leq 1/2$ and $x_t'[1] \geq 0$, we can lower bound the drift in the $e_1$ direction by

$$x_t'[1] + \frac{e^{-2(T-t)}}{\sigma^2}(Rv[1]/2 - \langle x_t', v \rangle).$$

Then for all $t \leq T^*$, by integrating we conclude that

$$x_t'[1] \geq x_0'[1]e^t + \frac{(1-e^{-t})e^{-2(T-t)}}{4\sigma^2} \cdot Rv[1].$$

So provided that $T^* > T_1$, by our choice of $T_1$ above we have that $\tanh(x_{T_1}'[1]) > 1/2$.

Next, for all $T_1 \leq t \leq T^*$, we will lower bound the drift in the $e_1$ direction by $x_t'[1]$, yielding the naive bound of $x_t'[1] \geq x_{T_1}'e^{t-T_1}$. Provided that $T^* > T_1'$, by our choice of $T_1'$ above we have that $\tanh(x_{T_1'}'[1]) \geq 1 - \epsilon$.

Next, let us consider the drift in the $v$ direction, which can be upper bounded by

$$\left(1 - \frac{e^{-2(T-t)}}{\sigma^2}\right)\langle x_t', v \rangle + Re^{-2(T-t)}v[1] \cdot (1 + 1/\sigma^2).$$

Furthermore, for all $t \leq T_2$, the drift in the $v$ direction is nonnegative. So by integrating the above upper bound, we conclude that for $\delta'$ defined as in Eq. (6), we have

$$\langle x'_{T'_1}, v \rangle \leq e^{-\frac{\delta' + e^{-2T}}{2\sigma^2}} \sqrt{\delta'} \cdot e^T \langle x'_0, v \rangle + \sqrt{\frac{\pi}{2}} Rv[1] \sqrt{\delta'} (\sigma + \sigma^{-1}) \left( \mathrm{erfi}\left( \sqrt{\frac{\delta'}{2\sigma^2}} \right) - \mathrm{erfi}\left( \frac{e^{-T}}{\sigma\sqrt{2}} \right) \right)$$

$$\lesssim \sigma \sqrt{\frac{\log(1/\epsilon)}{Rv[1]}} \cdot \langle x_0^{\mathsf{MDPS}}, v \rangle + \log(1/\epsilon) \leq c'\sigma \sqrt{\log(1/\epsilon) Rv[1]} + \log(1/\epsilon), \tag{10}$$

for some small constant $c > 0$, where $\mathrm{erfi}(\cdot)$ denotes the imaginary error function, and where in the second step we used that $\mathrm{erfi}(\sqrt{\frac{\delta'}{2\sigma^2}}) \lesssim \sqrt{\frac{\delta'}{2\sigma^2}}$ as well as the definition of $x'_0 = e^{-T} x_0^{\mathsf{MDPS}}$, and in the third step we used the assumption that $\langle x_0^{\mathsf{MDPS}}, v \rangle \leq cRv[1]$ for sufficiently small absolute constant $c$.

### D.4.2 STAGE 2: $t = T'_1$ TO $t = T_2 = T - \log(1/\sigma)$

Next we show that from time $T'_1$ to time $T_2$, $\langle x'_t, v \rangle$ steadily increases to $O(R\sigma)$.

First note that by definition of $T_2$, both $x'_t[1]$ and $\langle x'_t, v \rangle$ are nondecreasing for $0 \leq t \leq T_2$. Furthermore, at the end of the previous stage, we saw that $\tanh(x'_{T'_1}[1]) \geq 1 - \epsilon$, so $\tanh(x'_t[1]) \geq 1 - \epsilon$ for all $T'_1 \leq t \leq T_2$. As a result, the drift in the $v$ direction is upper bounded by

$$\left( 1 - \frac{e^{-2(T-t)}}{\sigma^2} \right) \langle x'_t, v \rangle + \left( e^{-2(T-t)} + \frac{e^{-2(T-t)}(e^{-2(T-t)} + 4\sigma^2)}{\sigma^2} \right) Rv[1].$$

Integrating this, we conclude that

$$\langle x'_{T_2}, v \rangle \leq Rv[1]\sigma^2 + e^{\delta'/2\sigma^2 - 1/2} \cdot \left( \frac{\sigma}{\sqrt{\delta'}} \langle x'_{T'_1}, v \rangle - Rv[1]\sigma\sqrt{\delta'} \right)$$

$$+ 2\sqrt{2\pi\delta'} Rv[1]\sigma \left( \mathrm{erfi}(1/\sqrt{2}) - \mathrm{erfi}(\sqrt{\delta/2\sigma^2}) \right)$$

$$\leq c'' Rv[1]\sigma, \tag{11}$$

where $\delta'$ is defined in Eq. (6), and in the second step we used the bound on $\langle x'_{T'_1}, v \rangle$ from Eq. (10), and $c''$ is a small absolute constant depending on $c$ in the assumed bound of $\langle x_0^{\mathsf{MDPS}}, v \rangle \leq cRv[1]$.

As $\langle x'_t, v \rangle$ has been nondecreasing up to this point, and the final bound in Eq. (11) is at most $\epsilon Rv[1]/2$ by our choice of $\epsilon$ in Eq. (9), we conclude that our running assumption that time $T^*$ happens after Stages 1 and 2 is valid.

### D.4.3 STAGE 3: $t = T_2$ TO $t = T$

We now complete the analysis of the trajectory by considering $t \geq T_2$. Define $\epsilon_t \triangleq 1 - \tanh(Rx'_t[1])$ so that the drift in the $v$ direction can be written as

$$\left\langle \frac{\mathrm{d}x'_t}{\mathrm{d}t}, v \right\rangle = \left( 1 - \frac{e^{-2(T-t)}}{\sigma^2} \right) \langle x'_t, v \rangle + \left( e^{-2(T-t)} + \frac{e^{-4(T-t)}}{\sigma^2} \right) Rv[1] + \Delta_t$$

for

$$\Delta_t = \left( \frac{e^{-2(T-t)}(1 - e^{-2(T-t)})}{\sigma^2} - e^{-2(T-t)} \right) Rv[1]\epsilon_t.$$

Writing $t = T_2 + s$, we can express the above as

$$\left\langle \frac{\mathrm{d}x'_{T_2+s}}{\mathrm{d}s}, v \right\rangle = (1 - e^{2s}) \langle x'_{T_2+s}, v \rangle + (e^{2s} + e^{4s}) Rv[1]\sigma^2 + \Delta_{T_2+s}$$

and

$$\Delta_{T_2+s} = (1 - 2\sigma^2 e^{2s}) Rv[1]\epsilon_{T_2+s}$$

We will regard this as a perturbed version of the ODE

$$\frac{\mathrm{d}Z_s}{\mathrm{d}s} = (1 - e^{2s}) Z_s + (e^{2s} + e^{4s}) Rv[1]\sigma^2, \tag{12}$$

which admits the solution

$$Z_s = e^{2s} Rv[1]\sigma^2 + e^{1/2 - e^{2s}/2 + s}(Z_0 - Rv[1]\sigma^2) \triangleq f_s(Z_0).$$

Note that $|f'_s(Z_0)| = e^{1/2 - e^{2s}/2 + s}$, and $\int_0^\infty e^{1/2 - e^{2s}/2 + s} \, ds \leq 2/3$. So by the Alekseev-Gröbner formula, if $Z_0 = \langle x'_{T_2}, v\rangle$,

$$\left| Z_s - \langle x'_{T_2+s}, v\rangle \right| = \left| \int_0^s f'_r(Z_r) \cdot \Delta_r \, dr \right| \leq \frac{2}{3} \sup_{0 \leq r \leq s} |\Delta_r|.$$

Let $T_3(\xi)$ denote the smallest time $T_2 \leq t \leq T$ for which $\tanh(x'_t[1]) \leq 1 - \xi$; if no such $t$ exists, let $T_3(\xi) \triangleq \infty$. Then for any $0 \leq s \leq T_3(\xi) - T_2$, we have

$$\left| \langle x'_{T_2+s}, v\rangle - \left( e^{2s} Rv[1]\sigma^2 + e^{1/2 - e^{2s}/2 + s}(\langle x'_{T_2}, v\rangle - Rv[1]\sigma^2) \right) \right| \leq \frac{2}{3} Rv[1]\xi \qquad (13)$$

Let us now track the evolution of $x'_{T_2+s}[1]$. We can always lower bound the drift in the $e_1$ direction by

$$x'_{T_2+s}[1] + (1 - \xi)R\sigma^2 e^{2s} + e^{2s}v[1](Rv[1]\xi + 0.99Rv[1]e^{2s}\sigma^2 - \langle x'_{T_2+s}, v\rangle)$$
$$\geq x'_{T_2+s}[1] + (1 - v[1])R\sigma^2 e^{2s} - \xi R\sigma^2 e^{2s} + 0.99R\sigma^2 e^{4s}v[1]^2$$
$$- e^{1/2 - e^{2s}/2 + 3s}c''Rv[1]^2\sigma + \frac{1}{3}e^{2s}Rv[1]^2\xi,$$

where in the second step we used Eq. (13) and the bound Eq. (11) established in Stage 2. In particular, for $\xi = C\sigma$ for sufficiently large constant $C$ relative to $c''$, we see that the above drift is nonnegative, implying that $\tanh(x'_t[1])$ can never go below $1 - O(\sigma)$. In particular, for this choice of $\xi$, $T_3(\xi) = \infty$.

We have thus established that Eq. (13) holds for all times $0 \leq s \leq T - T_2$. In particular, at the end of the trajectory, i.e. when $s = T - T_2 = \log(1/\sigma)$, we can bound the final measurement loss by

$$\left| \langle x'_T, v\rangle - Rv[1] \right| \lesssim e^{-\frac{1}{2\sigma^2}} \cdot Rv[1]\sigma + Rv[1]\sigma \log(1/\sigma) \lesssim Rv[1]\sigma \log(1/\sigma).$$

# E  ADDITIONAL EXPERIMENTAL RESULTS

In this section we report additional experimental results. In Section E.1, we report similar improvements using REGUIDANCE on CIFAR-10 and provide visual examples comparing our method to baselines. In Section E.2, we provide additional details for our experiments with superresolution on ImageNet as well as visual examples of outputs of REGUIDANCE for both box in-painting and superresolution. In Section E.3.1 we explore how the choice of latent affects the behavior of REGUIDANCE and provide visualizations for the space of good latents. Finally, in Section E.3.2, we validate the theoretical result of Theorem 2 by showing that empirically, REGUIDANCE using the SDE performs considerably worse compared to REGUIDANCE using the ODE.

## E.1  RESULTS ON CIFAR-10

### E.1.1  EMPIRICAL RESULTS

To validate REGUIDANCE on an alternative dataset, we take our top performing baseline (DAPS) and run our experiments (baseline with and without REGUIDANCE) on the CIFAR-10 dataset Krizhevsky et al. (2009), consisting of $32 \times 32$ images. We use the unconditional $32 \times 32$ diffusion model from Dhariwal & Nichol (2021) as the base model. As before, the inverse tasks are different scales of box-inpainting and superresolution: now, small inpainting denotes a $16 \times 16$ region masked out from the original image, while in large inpainting, the masked region dimensions are $23 \times 23$. All other settings with respect to evaluation are the same as with ImageNet. We report the results below.

As shown in Tables 3 and 4, the strong boost in performance offered by REGUIDANCE transfers to CIFAR 10. We see a near universal boost in performance across all tasks and metrics, especially seeing strong boosts in realism as quantified by the CMMD score.

| Method | Small Inpainting | | Large Inpainting | |
|--------|--------|--------|--------|--------|
| | LPIPS ↓ | CMMD ↓ | LPIPS ↓ | CMMD ↓ |
| DAPS | 0.147 | 0.519 | 0.300 | 0.636 |
| DAPS + REGUIDANCE | 0.151 | **0.373** | **0.298** | **0.560** |

Table 3: Experimental results for inpainting. Bold numbers show improvements over baseline, and underlined values mark the best result (not including last row which uses knowledge of ground truth).

| Method | Small Superresolution | | Large Superresolution | |
|--------|--------|--------|--------|--------|
| | LPIPS ↓ | CMMD ↓ | LPIPS ↓ | CMMD ↓ |
| DAPS | 0.392 | 0.591 | 0.516 | 0.859 |
| DAPS + REGUIDANCE | **0.383** | **0.569** | **0.509** | **0.669** |

Table 4: Experimental results for superresolution. Bold numbers show improvements over baseline, and underlined values mark the best result (not including last row which uses knowledge of ground truth).

### E.1.2 VISUAL EXAMPLES

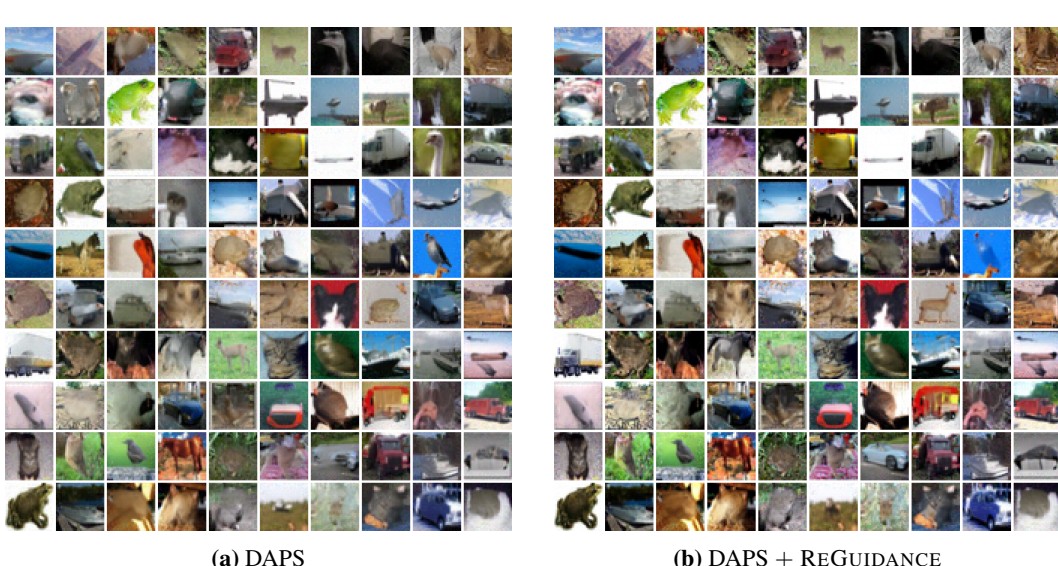

**(a)** DAPS  **(b)** DAPS + REGUIDANCE

**Figure 6:** Validation set generations for large inpainting on CIFAR-10.

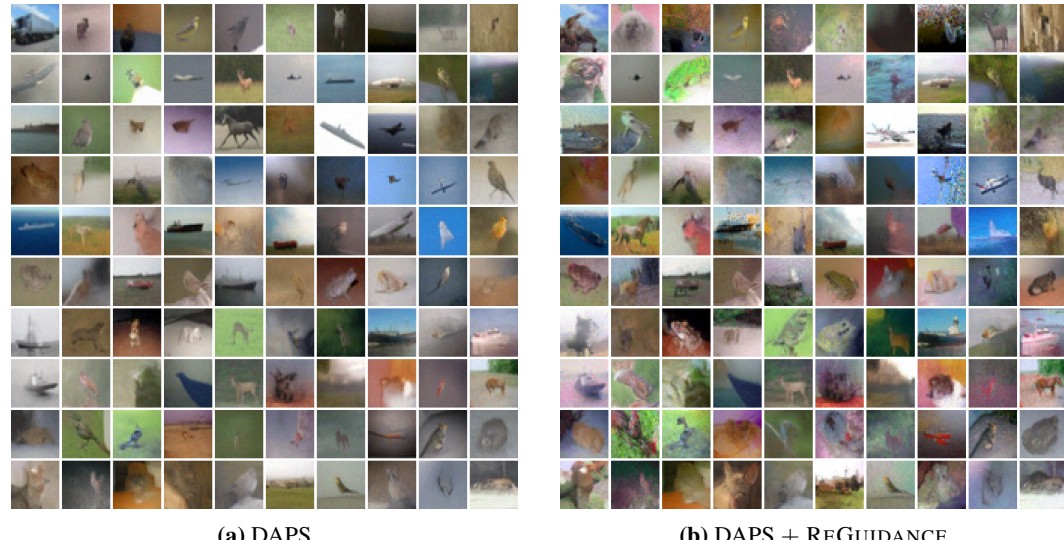

|  (a) DAPS | (b) DAPS + REGUIDANCE |

**Figure 7:** Validation set generations for large superresolution on CIFAR-10.

### E.2 ADDITIONAL RESULTS ON IMAGENET

#### E.2.1 SUPERRESOLUTION

| Method | Small Superresolution | | Large Superresolution | |
|---|---|---|---|---|
| | LPIPS ↓ | CMMD ↓ | LPIPS ↓ | CMMD ↓ |
| DAPS | 0.410 | 1.257 | 0.545 | 2.114 |
| DAPS + REGUIDANCE | 0.510 | 1.323 | 0.590 | **1.429** |
| DAPS + REGUIDANCE with ReNoise | **0.404** | **1.186** | **0.496** | 1.582 |
| DDRM | 0.393 | 1.232 | 0.511 | 1.549 |
| DDRM + REGUIDANCE | 0.405 | 1.435 | 0.530 | 1.971 |
| DDRM + REGUIDANCE with ReNoise | **0.391** | **1.221** | **0.505** | 1.704 |
| DPS | 0.457 | 0.566 | 0.523 | 0.576 |
| DPS + REGUIDANCE | **0.433** | 0.655 | **0.493** | 0.705 |
| Ground Truth + REGUIDANCE | 0.314 | 0.813 | 0.222 | 0.407 |

Table 5: Experimental results for superresolution. Bold numbers show improvements over baseline, and underlined values mark the best result (not including last row which uses knowledge of ground truth).

From Table 5, we note that REGUIDANCE helps boost reward in realism in some cases, while leaving the other metrics relatively unchanged / slightly worse. We note that performance using the ground truth latents of the original images have considerably better performance in reward and realism, suggesting that the candidate latents offered by the baselines might not be strong initializations (as reflected in the requirements of Theorems 1 and 3). Indeed, we observe that this could be due to both the baseline samples as well as due to discretization error in the inversion process using the reverse probability flow ODE. We show evidence of the latter in Figure 8, where running the deterministic diffusion ODE after inverting a candidate image does not quite return to said image, introducing noisy artifacts that are magnified by REGUIDANCE. These artifacts contribute to deteriorated performance (especially in the $8\times$ superresolution regime). To reduce the discretization error due to the inversion process, we employ a stronger inversion technique from ReNoise Garibi et al. (2024) and report our results in Table 5. We observe that a stronger inversion technique substantially improves the performance, and we leave the investigation of more advanced inversion techniques for future work.

| Method | Motion Deblurring | | Nonlinear Deblurring | |
|---|---|---|---|---|
| | LPIPS ↓ | CMMD ↓ | LPIPS ↓ | CMMD ↓ |
| DAPS | 0.667 | 2.457 | 0.572 | 1.651 |
| DAPS + REGUIDANCE | **0.560** | **1.736** | **0.474** | **0.876** |

Table 6: Experimental results for Motion and Nonlinear Deblurring. Bold numbers shows the best result.

### E.2.2 MORE VISUAL EXAMPLES

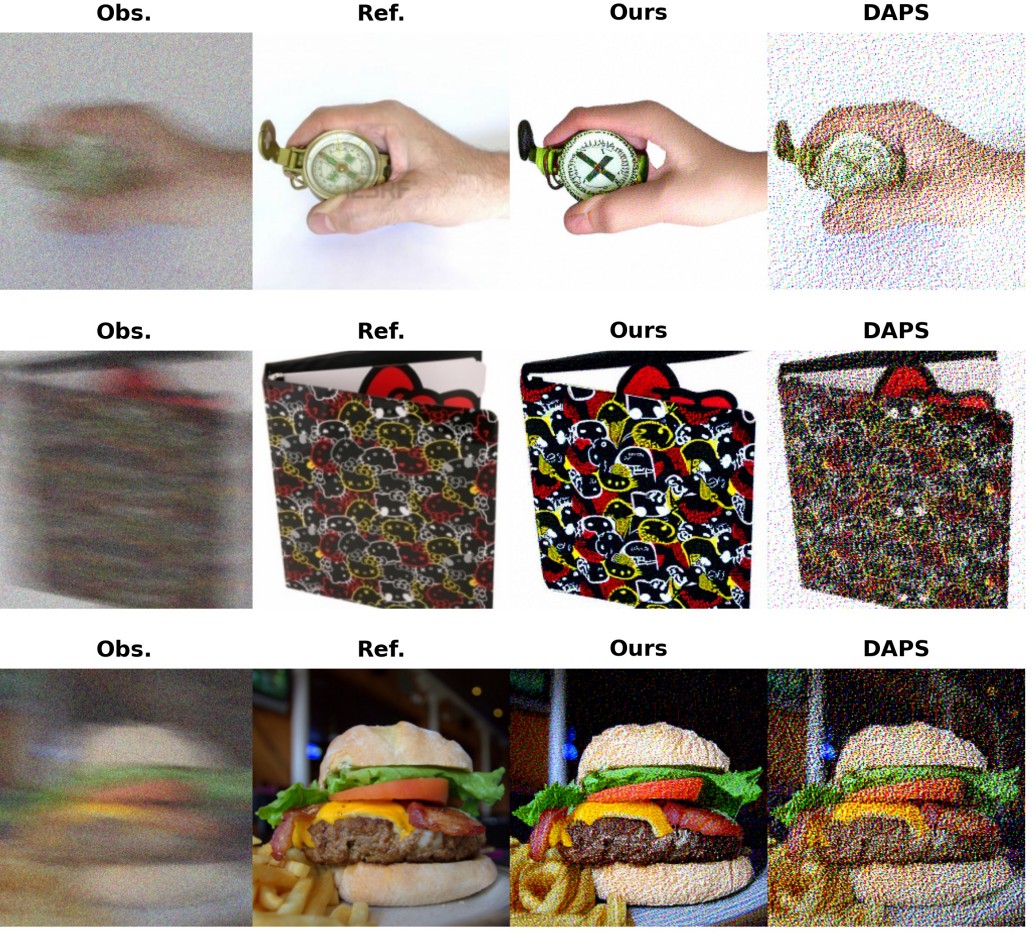

**Figure 9:** Hard motion deblurring.

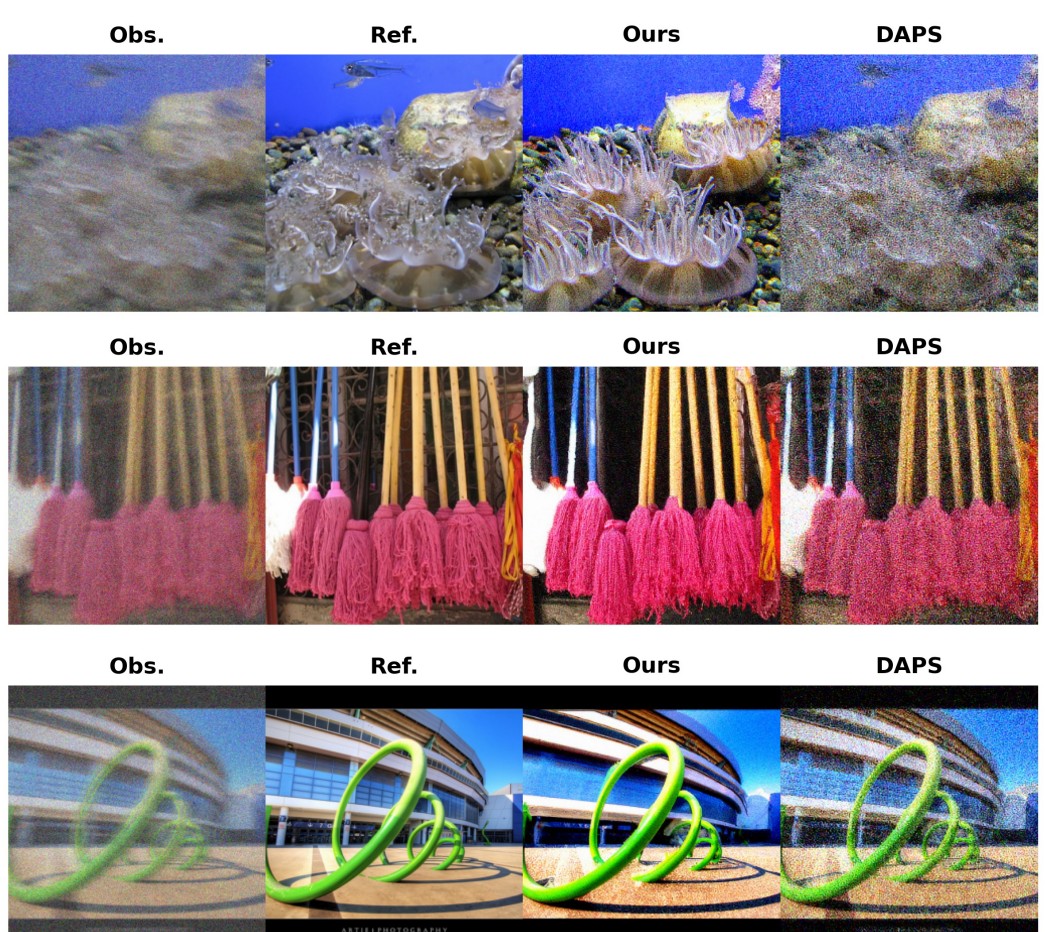

**Figure 10:** Hard nonlinear deblurring.

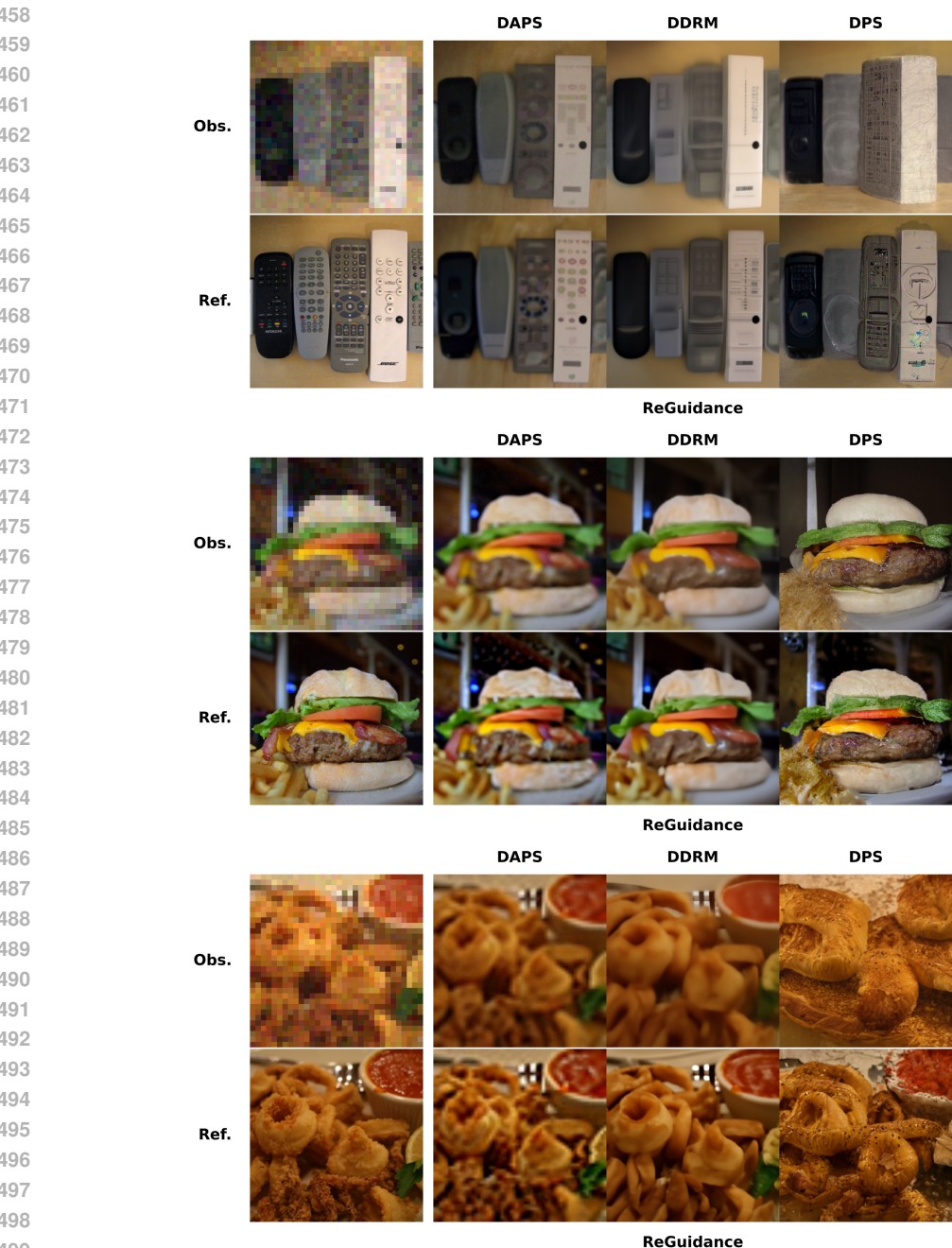

**Figure 11:** Small superresolution.

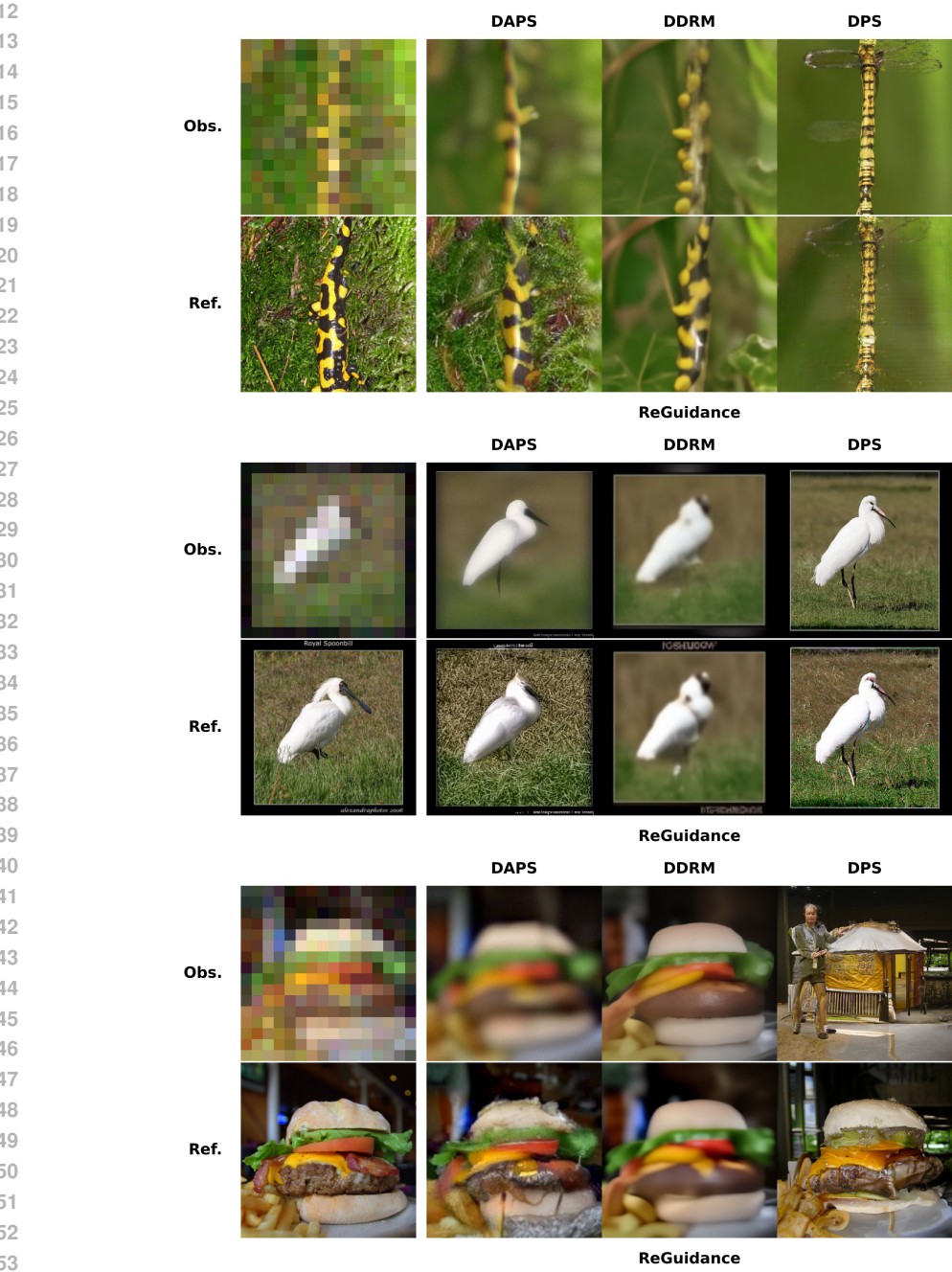

**Figure 12:** Large superresolution.

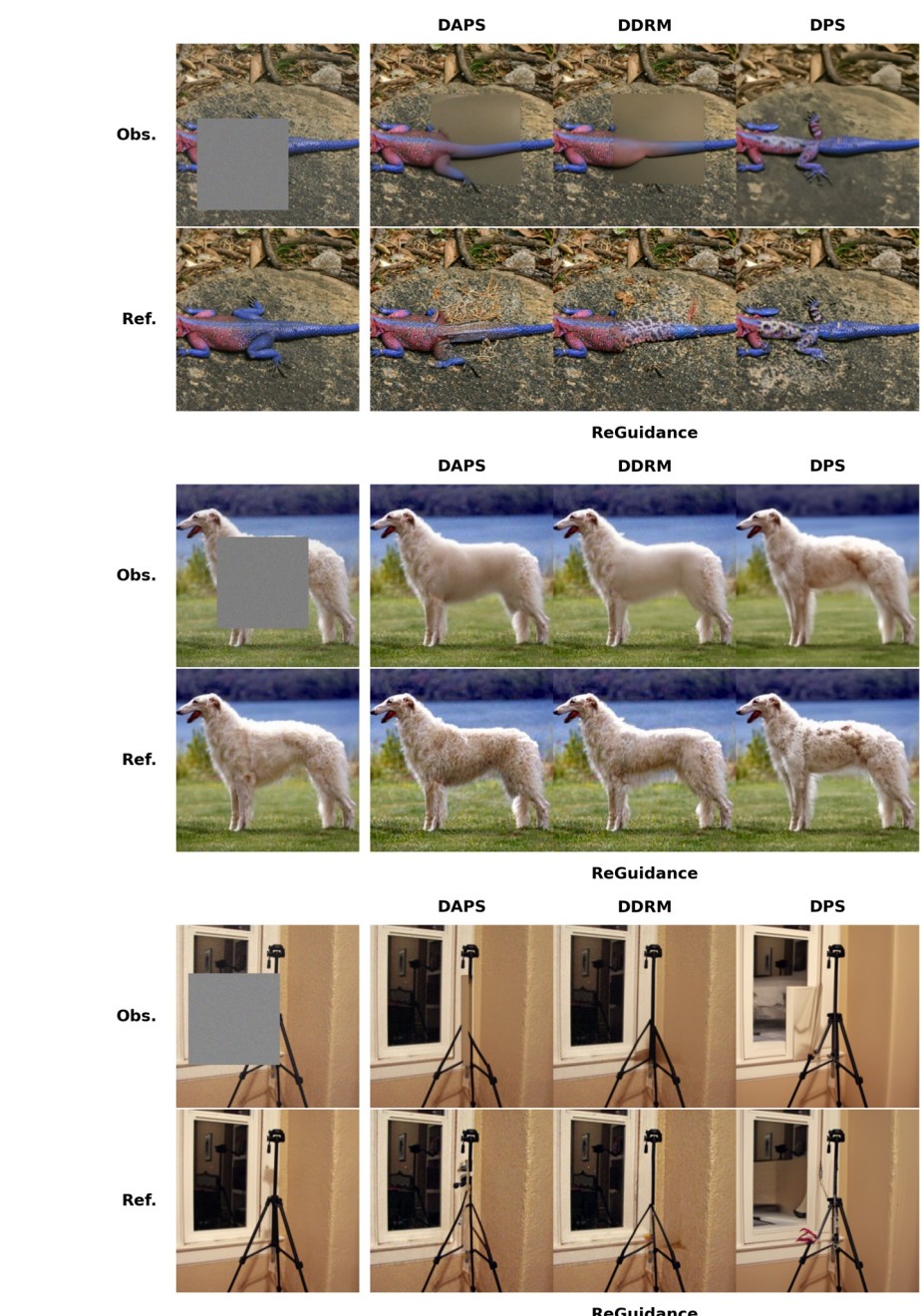

**Figure 13:** Small box-inpainting.

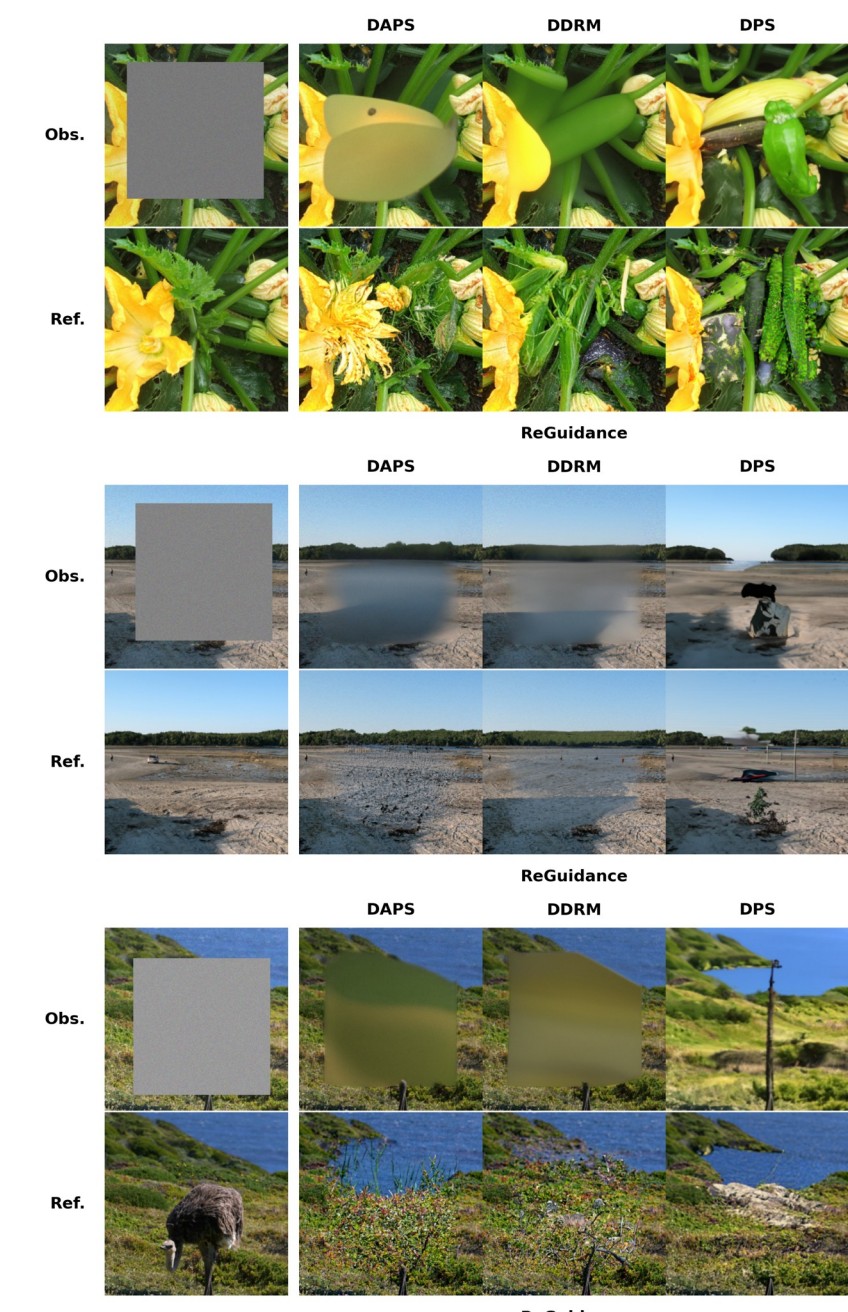

**Figure 14:** Large box-inpainting.

### E.3 QUALITATIVE BEHAVIOR

Here we describe further experiments to develop qualitative understanding for the behavior of REGUIDANCEas outlined in Section 4.3.

#### E.3.1 STRUCTURE OF LATENT INITIALIZATIONS FOR REGUIDANCE

We notice the space of good initial latents is *disconnected*, allowing many possible sample initializations to provide *strong* and *diverse* quality boosts. While a small $L_2$ ball of latents around good

latents also provide generally strong reconstructed samples, *it is not true* that all good latents are "neighbors" of each other.

We provide a qualitative experiment to test these assertions. For DAPS, DDRM, and DPS, we take their candidate solutions to a large inpainting problem (see Figure 3), along with the original ground truth image. Firstly, inverting these four images, we notice that the pairwise distances between the resulting latents is quite large: if $d$ is the dimensionality of the latents, then the average distance between two random standard normal latents is approximately $\sim 2\sqrt{d} = \ell$. Given the latents have shape $[3, 256, 256]$, we find that the pairwise distances between these latents lie in

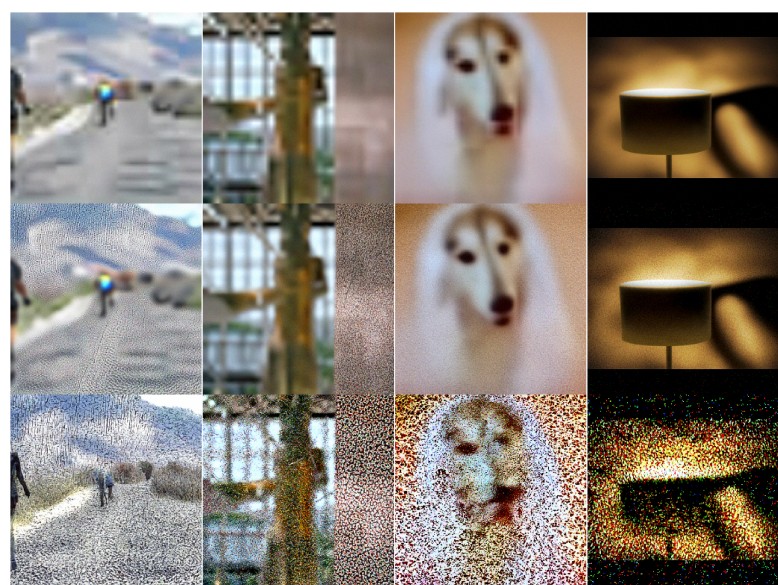

**Figure 8:** First row: candidate solution generated by DAPS; second row: image generated by running unconditional ODE from inverted ground truth latent; third row: image generated by REGUIDANCE with this latent. Second row does not quite give the identity map, and artifacts introduced by discretization error are magnified with REGUIDANCE. First two columns: $8\times$ superresolution; last two: $16\times$.

the interval $[0.5\ell, 0.7\ell]$. In particular, these "good" latents are quite far apart in space. In Figure 15, we see that perturbing these latents by a standard normal vector with standard deviation $0.1$ and applying REGUIDANCE results in realistic images quite similar to those generated from REGUIDANCE applied on the non-perturbed latents, regardless of the random seed. In particular, the good latents are robust up to a radius of $\sim 0.1\sqrt{d}$. Beyond this radius, e.g., interpolating two subsequent latents and then applying REGUIDANCE, the resulting images are much less realistic, and often contain visible defects.

This suggests that the space of good latents for REGUIDANCE is *disconnected*, with a small neighborhood around each good latent providing highly similar images, while distinct neighborhoods lead to *diverse* but *strong* solutions to the target inverse problem. In particular, good latents are not necessarily "neighbors" of each other, but exist in disjoint neighborhoods within initialization space.

### E.3.2 REGUIDANCE WITH SDE VS. ODE

These strong initializations rely on the *deterministic* sampler for REGUIDANCE. We demonstrate that REGUIDANCE with a stochastic DPS sampler (SDE) loses some of this benefit that a strong initialization provides. Using a classic unconditional SDE sampler for diffusion models is known to be memoryless Domingo-Enrich et al. (2025), in the sense that the generated sample does not depend on the initial latent. Adding Brownian motion to DPS similarly weakens dependency on the latent, deteriorating performance as we now show.

To implement REGUIDANCE with a stochastic sampler, we use DPS with DDIM but now with noise parameter $\eta = 1.0$. In Table 7, we see REGUIDANCE with this DPS-SDE underperforms the baseline in measurement consistency and underperforms the DPS-ODE in realism. Figure 16 shows sample REGUIDANCE DPS-ODE images relative to using the DPS-SDE, stressing the disparity in performance seen in Table 7.

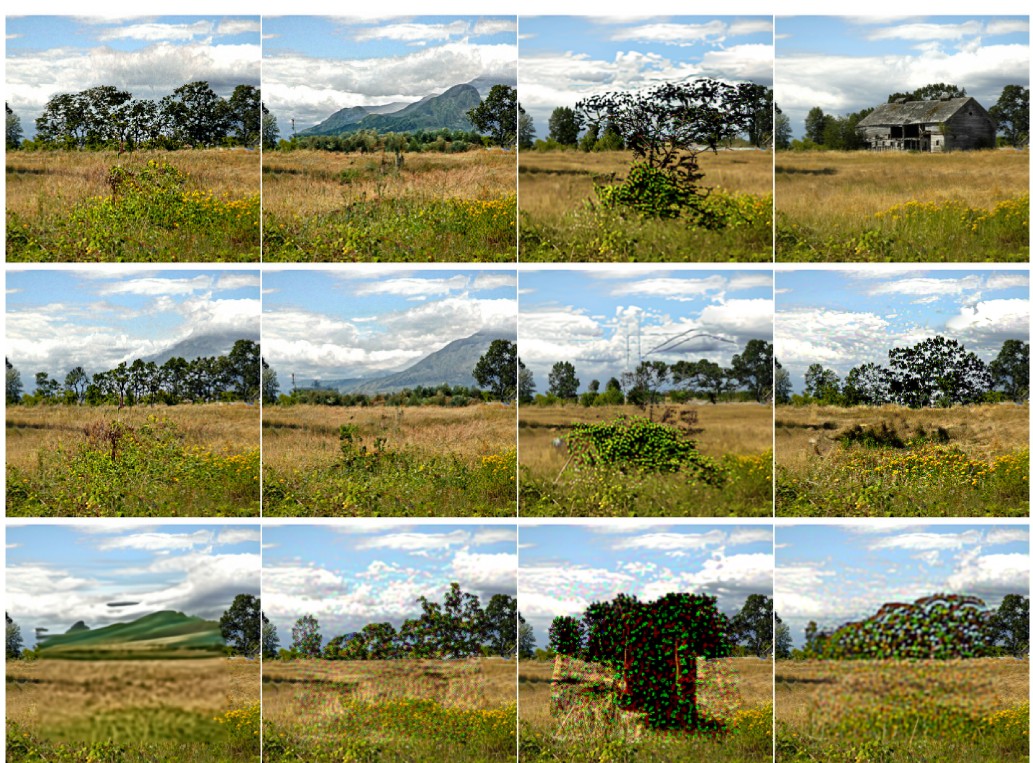

**Figure 15:** The first row displays solution images generated by REGUIDANCE using latent initializations from DAPS, DDRM, DPS, and the ground truth image respectively. The second row displays REGUIDANCE applied on these latent initializations perturbed by a random normal vector with variance 0.1. The final row displays REGUIDANCE applied on the current latent averaged with the initial latent of the next column of images (with wrap-around). This figure demonstrates that small neighborhoods around good latents still lead to realistic images, albeit with little variation, while across neighborhoods of good latents, diversity in infillings emerges.

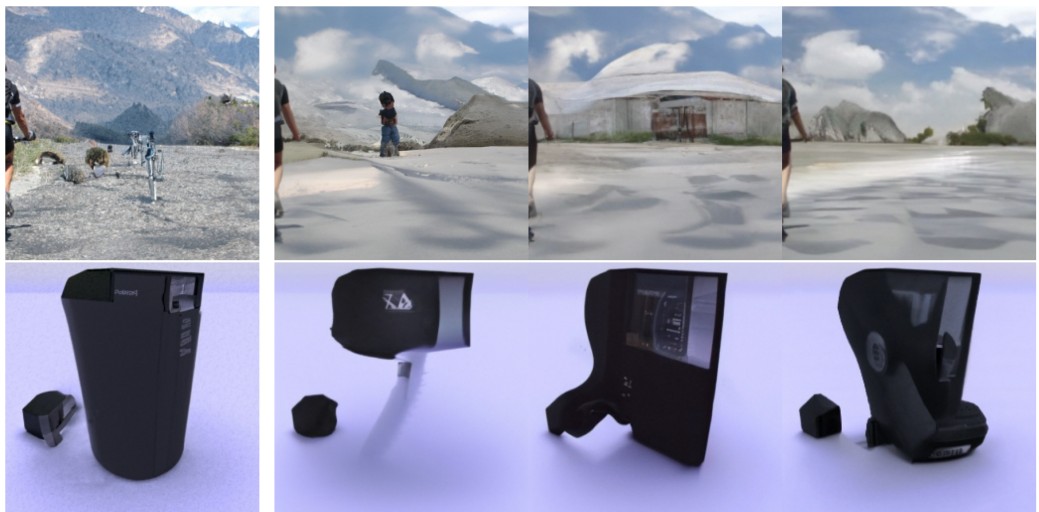

**Figure 16:** The first column gives REGUIDANCE with a deterministic DPS-ODE on DAPS candidate latents for large inpainting on ImageNet, and the subsequent three columns are different samples using REGUIDANCE with a stochastic DPS-SDE sampler. It is evident from the photos that both realism and measurement consistency are lost when using the stochastic REGUIDANCE sampler.

| Method | Large Inpainting (Box) | |
|---|---|---|
| | LPIPS ↓ | CMMD ↓ |
| DAPS | 0.418 | 1.103 |
| DAPS + REGUIDANCE (ODE) | **0.376** | **0.628** |
| DAPS + REGUIDANCE (SDE) | 0.470 | 0.815 |

Table 7: Comparing REGUIDANCE with the ODE vs the SDE for large box-inpainting on ImageNet. The SDE sampler results in much worse performance than both the baseline and the ODE with regards to consistency (LPIPS), and worse performance than the ODE with regards to realism (CMMD).

