# OpenReview forum: "ReGuidance: Diffusion Steering with Strong Latent Initializations Solves Hard Inverse Problems"
_ICLR.cc/2026/Conference — Submitted to ICLR 2026_

### Official Review · Reviewer_EPZz · 2025-10-26

**Soundness:** 3
**Presentation:** 2
**Contribution:** 2
**Rating:** 4
**Confidence:** 4

**Summary:**

This work proposes a training-free algorithm for diffusion-based inverse problem solver, with the key idea that inverse an initial recovery to latent representation through probability flow ODE, and then adopt the DPS from this initialization. Experiments show the effectiveness.

**Strengths:**

1. The author provide theoretical analysis of ReGuidance in improving reward and realism in Gaussian mixture toy setting;
2. Experiments show improvements over baselines on ImageNet;

**Weaknesses:**

1. The proofs are restricted to mixtures of Gaussians and linear inverse problems and far from real-world settings;
2. Only evaluated with ImageNet DDPM, and the evaluation set is quite small (100 samples);
3. There is no ablations on guidance strength, ODE steps, or computation trade-off;
4. The baselines are old, it is suggested to add some new strong baselines;

**Questions:**

1. It is unclear whether the reverse ODE process is stable or sensitive to score errors in used diffusion models, also, how sensitive are the improvements to the initial reconstruction quality, Does ReGuidance fail if the ODE reverse results is far from the manifold?
2. How does runtime scale with diffusion steps compared to DPS/DAPS?

---

> ### Author Response · Authors · 2025-11-20
> **Part I: Added Results**
>
> We thank the reviewer for their questions and comments! We address the raised concerns below:
>
> **The proofs are restricted to mixtures of Gaussians and linear inverse problems and far from real-world settings.**
>
> We hope the following context will help calibrate the reviewer’s expectations regarding theoretical results in the literature on diffusion guidance, which is largely dominated by heuristic arguments.
>
> Diffusion posterior sampling targets an algorithmic problem which can be **computationally hard even in the average case** (see [1]). As a result, it has been extremely tricky to identify settings where an end-to-end analysis of a heuristic used in practice is feasible. Indeed, current empirical works restrict their theoretical scope to strongly simplified models of the data. We note that **even among theoretical works on diffusion guidance which have appeared at the main ML conferences recently, it is standard to focus on Gaussian mixtures** (see [2] and [3]). In the particular setting of gradient guidance (DPS) we focus on in this submission, ours is the *first end-to-end theoretical guarantee* completely characterizing the sampling dynamics of the *empirically common algorithm of reward gradient steering*, for any model of data.
>
> **Only evaluated with ImageNet DDPM, and the evaluation set is quite small (100 samples).**
>
> We note that we evaluate on the CIFAR dataset as well in the Appendix (see Appendix E.1). In terms of the evaluation set size, the LPIPS/CMMD scores have low sample complexity and saturate at around 100 samples: in other words, the scores are similar whether a set of 100 or 1000 is used.
>
> To illustrate this point, we run our large inpainting evaluation for ReGuidance on 1000 samples, and report LPIPS/CMMD in conjunction with the 100 samples results:
>
> | Method                         | LPIPS ↓        | CMMD ↓        |
> |--------------------------------|----------------|----------------|
> | DAPS (100 / 1k)            | 0.418 / 0.424  | 1.103 / 1.036  |
> | DAPS + ReGuidance (100 / 1k) | **0.376 / 0.382**  | **0.628 / 0.675**  |
>
> As the table illustrates, the scores between 100 and 1k sized eval sets are very similar, and again, owes to the sample efficiency of the metrics we use.
>
> **There is no ablations on guidance strength, ODE steps, or computation trade-off.**
>
> In terms of ablations, similar to the original DPS paper (Appendix C [4]) we find that within an interval ($\pm 0.02$) of our guidance scales, performance differences are roughly negligible (on average, LPIPS and CMMD vary on the order of $0.01$). Setting the guidance scale too high introduces image artifacts such as grated patterns or oversaturation, and setting the scale too low recreates the candidate solution (i.e. reproduces the inverted baseline image). We will add this discussion to the final version of the paper.
>
> We present a more detailed analysis of the runtime tradeoffs with the baselines in terms of the number of steps below in response to a related question.
>
>
> **The baselines are old, it is suggested to add some new strong baselines.**
>
> We report results for DMPlug on large inpainting, a recent strong MAP-estimation method ([5]):
>
> | Method                   | LPIPS ↓ | CMMD ↓ |
> |--------------------------|---------|--------|
> | DMPlug               | 0.542   | 1.430  |
> | DMPlug + ReGuidance  | **0.443**   | **1.352**  |
>
> These preliminary results should demonstrate that even the most recent methods do not perform well on hard reward/inverse problem regimes, and that ReGuidance is a step towards meaningfully improving performance on such difficult tasks.
>
> **Sensitivity to score errors and reconstruction quality**
>
> Empirically, since score errors are inherent to trained diffusion models, the reverse ODE process in practice is stable to these score errors. Moreover, our theorems imply that discretization errors in inversion ODE can be tolerated, since the condition for strong initializations is a distance function in initialization space. As the results demonstrate, the bar for initial reconstruction quality is not particularly high either (see Fig. 1, 9, 10), so ReGuidance improvements are not highly sensitive to initial quality.
> In fact, the candidate solutions are not necessarily even close to the image manifold; ReGuidance (both theoretically and empirically) still boosts the solution closer to the image manifold (theory: Theorem 3 improving likelihood under the data distribution + empirically: CMMD score improvements).
>
> ----------------------------------------------------------
> **References**
>
> [1]: Bruna-Han 2024. “Posterior Sampling with Denoising Oracles via Tilted Transport”
>
> [2]: Wu et. al. "Theoretical Insights for Diffusion Guidance" (ICML ‘24)
>
> [3]: Chidambaram et. al. "What does guidance do?" (NeurIPS '24)
>
> [4]: Chung et. al. 2023. "Diffusion Posterior Sampling for General Noisy Inverse Problems"
>
> [5]: Wang et al. 2024. "DMPlug: A Plug-in Method for Solving Inverse Problems with Diffusion Models"

---

> ### Author Response · Authors · 2025-11-20
> **Part II: Runtime**
>
> **How does runtime scale with diffusion steps compared to DPS/DAPS?**
>
> In general, if the baseline method takes time B, and there are N diffusion steps, ReGuidance has runtime $B$ + $N \cdot $ Inv + $N \cdot $ ReG, where Inv is the time per inversion step and ReG is the time per ReGuidance step. Empirically (for ImageNet 256x256), Inv ~ $0.06$ s, while ReG ~ $0.18$ s, since the DPS-ODE requires differentiating through the reward. For $N = 1000$, our runtime simplifies to $B + 4$ minutes. In comparison, note that DPS takes a time of $N \cdot $ ReG, while DAPS uses Langevin steps and takes time $N \cdot $ Inv + $2000 \cdot $ Langevin. In practice, using the numbers from above, DPS takes $3$ minutes while DAPS takes $1$ min.
>
> We will include this analysis in the final version of our paper!

---

### Official Review · Reviewer_ev17 · 2025-10-31

**Soundness:** 1
**Presentation:** 3
**Contribution:** 2
**Rating:** 4
**Confidence:** 4

**Summary:**

The paper introduces an algorithm called REGUIDANCE that improves the solution of existing diffusion-based plug-and-play inverse problem solvers. Specifically, given a solution $x$ of an existing diffusion-based inverse problem solving algorithm, REGUIDANCE proposes to run the probability flow ODE in reverse, i.e., starting from the clean image $x$ to generate an intermediate latent $x_T$ at time $T$ and then use this $x_T$ as an initialization to run the DPS method using the ODE formulation, i.e., DPS-ODE. Considering a standard toy case, the paper conducts a theoretical analysis and provides bounds on certain quantities, which were useful in inferring about the realism and measurement consistency of the solution returned by the REGUIDANCE procedure. Quantitative metrics such as LPIPS have been shown to improve when REGUIDANCE is applied on top of existing algorithms.

**Strengths:**

The proposed procedure, called REGUIDANCE, seems simple yet very effective in improving the solution returned by existing inverse problem solvers.  The approach is also modular, which implies that the procedure can be used on top of any existing solvers, which widens its impact.

In some toy settings, the claims about the solutions returned by REGUIDANCE are supported by a thorough analysis, which adds validity to the proposed method and indicates a principled nature of the approach.

Most inverse problem solvers can perform well for simple inverse problems, but they do really struggle with “hard inverse problems,” and yet this hasn’t been paid much attention to in the literature. I strongly agree and appreciate the authors’ choice of using “hard datasets” like ImageNet and CIFAR rather than FFHQ, etc., and also “hard inverse problems” such as extreme mask inpainting and extremely downsized super-resolution, etc. Improvements on such hard problems showcase the true effectiveness of REGUIDANCE in practice.

Though minor concerns exist, I find the paper well written, easy to read, and organized quite well. I find the roadmap in the appendix very helpful, along with the informal remarks and the simplified explanations in the main text and appendix. I appreciate the authors’ thoughtfulness in this regard.

**Weaknesses:**

Though the paper presents an interesting find, I believe it has the following weaknesses regarding the plausibility of conclusions drawn from the theoretical analysis, and about where the actual effectiveness of the method stems from. These concerns further highlight the need for more experiments and ablations (some suggested below), which I believe are critical to address before the paper can be recommended for acceptance.

Q1. Remark 1 (line 329) mentions that REGUIDANCE improves the posterior likelihood. But prior works such as DMAP[1] have shown that the DPS procedure itself encourages MAP solutions, i.e., boosts posterior likelihood. In light of this fact, I believe the true effectiveness of REGUIDANCE arises from the fact that the DPS-ODE component of REGUIDANCE encourages MAP solution by default, i.e., irrespective of initialization. This might also explain why it has to be DPS-ODE and not some other posterior sampling algorithm?  Also, from remark 1, if  REGUIDANCE’s effectiveness is because it improves the posterior likelihood, then can one also expect it to perform well if DPS-ODE is replaced with other MAP solvers? I understand that the authors seemed to be completely unaware of the work DMAP[1], as it hasn’t been cited in the paper. From the above perspective, I find REGUIDANCE less novel than posed in the paper (of course, this doesn’t overshadow the other contributions of the paper). Still, I see REGUIDANCE as yet another improvement of DPS (like DMAP[1]) for MAP estimation. However, the most novel aspect of REGUIDANCE is that this improvement is based on good initializations and no procedural changes to DPS, unlike DMAP[1], which alters the DPS procedure slightly. I would ask the authors to clarify and justify their case if something is different from my understanding above.

Q2. The paper only considers posterior sampling algorithms such as DDRM, DPS, and DAPS, but fails to consider MAP solvers such as DMPlug[2], MAP-GA[3], ProjDiff[4], etc. Again, from remark 1, if the ultimate reason behind REGUIDANCE’s effectiveness is that it returns high-likelihood sample of the posterior $p(x|y)$, then it should absolutely be (1) thoroughly compared to existing MAP-based methods, such as checking whether REGUIDANCE+DDRM/DPS/DAPS solutions perform comparably to the solutions returned by MAP methods, and (2) if (1) holds, then it should be checked how much improvement REGUIDANCE+MAP solver offers over the vanilla MAP solver solution.


Q3. Concerning the point above, I understand that MAP solvers can be computationally expensive than posterior sampling; however, the REGUIDANCE procedure also seems to be quite expensive as it first needs to run the original posterior sampler, then run PF ODE to generate the latents, and finally run DPS-ODE. However, no mention of the computational efficiency of REGUIDANCE was discussed. Also, no mention of NFEs, the noise schedules, and other hyperparameters for REGUIDANCE and baselines was made in the whole paper. A study on how the performance of REGUIDANCE depends on the hyperparameters of DPS, especially such as $\rho$, and NFEs, noise schedule, etc., is crucial and quite essential for empirical validity in my opinion.

Q4. Regarding Theorem 1 and Theorem 3, I find it difficult to understand the extent to which the conclusions apply beyond toy cases to the real case of Image inpainting. Especially, Theorem 3 holds if the initial solution $x$ is “sufficiently” close to the mode $z_1$? This is a very unrealistic assumption, for (1) this assumption clearly doesn’t hold in the case of the initial solutions returned by DDRM/DAPS/DPS for image inpainting. (2) If the initial solution itself satisfies measurement and is closer to a mode, i.e., a highly likely sample of the posterior, then REGUIDANCE might further push the solution closer to the mode, but one would expect the returned solution of REGUIDANCE to be perceptually similar to the initial solution, which may result in marginal improvements of LPIPS. I’d recommend an empirical verification of this if the authors would consider MAP solvers, as mentioned in point 2 above.


Q5. Another crucial aspect of Theorems 1 and 3 concerning why initializing (for DPS-ODE) at a higher $T$ is better:  Theorem 3 bounds say, if $T$ is large, then $x_{T}^{DPS}$ has more realism, (but from Theorem 1 bound, if $T$ is large, $x_{T}^{DPS}$ has low reconstruction error, but it also says that the unobserved pixels remain closer to the initial solution $x$.) this clearly implies that it is heavily based on the fact that $x$ is already very close to a realistic sample. Since $x$ is not already a realistic sample in practice, I find the argument for a higher $T$ quite unconvincing. In my opinion, this aspect, like point 4 above, needs an empirical validation by considering intermediate time initializations (with the same NFEs, however) and not just max $T$ initialization.


Q6. With intermediate initialization in point 5 above, I strongly recommend an ablation to check if REGUIDANCE can still improve the initial solution if the latent is generated with SDE instead of PF-ODE. I believe this aspect of why only the PF-ODE-generated latent has to be used is not discussed in the paper (i.e., why not SDE-generated latent + DPS-ODE with the same NFEs, but with intermediate time initializations, because if $T$ is large, since the latent becomes random, REGUIDANCE essentially becomes DPS).



[1] Xu et al. Rethinking Diffusion Posterior Sampling: From Conditional Score Estimator to Maximizing a Posterior
[2] Wang et al. DMPlug: A Plug-in Method for Solving Inverse Problems with Diffusion Models
[3] Gutha et al. Inverse Problems with Diffusion Models: A MAP Estimation Perspective
[4] Zhang et al. Unleashing the Denoising Capability of Diffusion Prior for Solving Inverse Problems


Minor weaknesses/clarifications:

Q7. The notation of $x_t$ can be made more clear. In Sec 2.1, $t$ always goes from 0 to T. The left and right arrows clarify whether we talk about the reverse or the forward process, but in some later parts of the text, such as line 203 (one would think $x_{0}^{DPS}$ to be a clean image) and line 290 (one would think $t=0$ in $v_{t}^{DPS}$ is at the start of the reverse process because of earlier notation in line 203). Although I find the paper very interesting, the notation inconsistencies make it difficult to read.


Q8. I’d appreciate it if the authors could share additional qualitative visualizations of imagenet examples over the large mask image inpainting task. From my own experience, I find that sometimes we can achieve better LPIPS if the reconstructed images have rich texture, but are not necessarily semantically meaningful (This can imply unrealistic samples getting better LPIPS. However, I understand this can also happen due to other reasons, such as diffusion models not being perfect, etc., so I’m not very critical about this.)

**Questions:**

Please see the weaknesses mentioned above.

---

> ### Author Response · Authors · 2025-11-20
> **Responses to Q1-Q4**
>
> We thank the reviewer for their immensely detailed and thoughtful comments! We address the concerns in order below:
>
> **Regarding comparisons to MAP solvers and the importance of initialization (Q1/Q2)**
>
> We argue that encouraging MAP is not doing the “heavy lifting” in our empirical results, but it is precisely the strong initializations coupled with the DPS-ODE jointly that lead to performant results. We use the deterministic version of DPS precisely to utilize the initialization information, as an SDE would destroy this information content. As our theorems demonstrate, gradient guidance (DPS) coupled with this initialization improves the realism/reward.
>
> To ablate for this, as the reviewer has suggested, we explore a MAP solver (DMPlug) as a baseline, which simply encourages MAP solutions irrespective of initialization. We found that DMPlug does not perform very well on the large inpainting task (worse than even DAPS without ReGuidance). The DMPlug per sample runtime for ImageNet was $40$ min per sample, compared to our runtime of $\leq 7$ min per sample.
>
> In fact, ReGuidance offers a performance boost on top of DMPlug, but this does not perform nearly as well as DMPlug+DAPS or DMPlug+DDRM, demonstrating that better initializations are crucial to the overall performance of ReGuidance, not just the MAP-tendencies of the DPS-ODE.
>
> | **Method**               | **LPIPS ↓** | **CMMD ↓** |
> |--------------------------|-------------|-------------|
> | DMPlug                   | 0.542       | 1.430       |
> | DMPlug + ReGuidance     | **0.443**       | **1.352**       |
> | DAPS                     | 0.418       | 1.103       |
> | DAPS + ReGuidance       | **0.376**       | **0.628**       |
>
>
> **Regarding computational efficiency/ablations of ReGuidance (Q3)**
>
> In general, if the baseline method takes time $B$, and there are $N$ diffusion steps, the total time for ReGuidance is $B$ + $N \cdot$ Inv + $N \cdot$ ReG, where Inv is the time per inversion step and ReG is the time per ReGuidance step. Empirically (for ImageNet 256x256), Inv ~ $0.06$ s, while ReG ~ $0.18$ s, since the DPS-ODE requires differentiating through the reward. For $N = 1000$, the ReGuidance has runtime $B + 4$ min, so methods like DDRM ($B = 20$ sec) and DAPS ($B = 1$ min) are reasonable at inference given the performance boost. In comparison, a MAP method like DMPlug ends up taking $40$ min per sample for ImageNet 256x256.
>
> In terms of ablations, similar to the original DPS paper (Appendix C [1]) we find that within an interval ($\pm 0.2$) of our guidance scales, performance differences are roughly negligible (on average, LPIPS and CMMD vary on the order of $0.01$). Setting the guidance scale too high introduces image artifacts such as grated patterns or oversaturation, and setting the scale too low recreates the candidate solution (i.e. reproduces the inverted baseline image). We will add this discussion to the final version of the paper.
>
> **Regarding sufficient closeness of the initialization (Q4)**
>
> Thanks for pointing out this confusing terminology in Theorem 3! We use “sufficiently close” as convenient language to express that a "good initialization" is within some distance of a ground truth latent, but the actual theorem does not actually demand very close proximity in initialization, as the experiments demonstrate. In Theorem 6 (formal version of Theorem 3), we present the mathematical condition for a “good initialization”, which appeals to the existence of a constant (that can be large) that defines a distance bound from a ground truth latent. Of course, in practice, we see that "good initializations" do not have to be terribly strong; i.e. the constant can indeed be large.
>
> --------------------------------------
> **References**
>
> [1]: Chung et. al. 2023. Diffusion Posterior Sampling for General Noisy Inverse Problems

---

> > ### Author Response · Authors · 2025-11-20
> > **Responses to Q5-Q8**
> >
> > **Regarding intermediate initializations (Q5/Q6)**
> >
> > Regarding the reviewer’s interpretation of Theorems 1 and 3 and the comment about large $T$, we would like to clarify a potential point of confusion. Theorem 1 **does not** say that the larger $T$ is, the closer the “unobserved/masked pixels” of $x^{DPS}_T$ (clean image) are to those of the initial reconstruction $x$. Theorem 1 only uses sufficiently large $T$ to ensure that the observed pixels become consistent with the measurement. In other words, $T$ ensures the inpainted measurement that **is observed** is consistent with the generated clean image from ReGuidance. This is how $T$ shows up in Theorem 3/6 as well: high $T$ ensures consistency with the observation (reward), but the mechanism for boosting realism (i.e. the unobserved pixels) is distinct and separate from the choice of $T$.
> >
> > Empirically, we can think of initializing at an intermediate timestep as starting with some partial “semantic” information of the input image (from the baseline), and then applying the DPS-ODE. For the emergence of information throughout the diffusion process, the middle timesteps (roughly $t=\frac{4T}{5}$ to $t=T$) correspond to high level semantic emergence, $t=\frac{T}{5}$ to $t=0$ is low-level refinement, and $t=T$ to $t=\frac{4T}{5}$ is a mode-selection phase. Initializing between $t=\frac{T}{5}$ and $t=0$ preserves the generated semantics of the baseline image, while initializing between $t=T$ and $t=\frac{4T}{5}$ is roughly equivalent to initializing at $t=T$.
> >
> > To the reviewer's suggestion for ablations, the middle phase is the most interesting in terms of a potential comparison. However, even on single-sample ablations, we found that inverting to medium-scale timesteps and then applying the DPS-ODE consistently introduced "large-scale semantic" artifacts that deteriorated the final image quality. A similar observation was found using the SDE (i.e. noising to an intermediate scale and then running ReGuidance). The generated "image semantics" at these intermediate timescales seem incompatible with imposing sudden gradient guidance at an intermediate stage.
> >
> > **Regarding notation (Q7)**
> >
> > We thank the reviewer for bringing this up! We will edit the surrounding discussion in the paper to clarify any confusion with notation regarding timesteps corresponding to the beginning vs. the end of the reverse process.
> >
> > **Regarding more qualitative visualizations (Q8)**
> >
> > The reviewer brings up a good point; we will include additional reconstructed images for large inpainting, but we remark that the CMMD score is trained on human preferences to discount “defects” like over-detailed but unrealistic textures that essentially amount to a complex noise distortion. Thus a higher CMMD score is also indicative of better realism.

---

### Official Review · Reviewer_Xb4y · 2025-10-31

**Soundness:** 4
**Presentation:** 3
**Contribution:** 3
**Rating:** 4
**Confidence:** 4

**Summary:**

This paper proposes a simple yet effective method to enhance diffusion-based inverse problem solving by leveraging strong latent initializations. The core idea is to take an initial reconstruction \(x\) from any baseline method, invert it via the unconditional probability flow ODE to obtain a latent $x_T^*$, and then run a deterministic DPS-ODE from this latent to produce an improved sample $\hat{x}$. Theoretically, the authors prove in Gaussian mixture models that REGUIDANCE boosts both reward (measurement consistency) and realism (data likelihood). Empirically, REGUIDANCE consistently improves state-of-the-art baselines (DDRM, DPS, DAPS) across challenging image restoration tasks (large inpainting, super-resolution, deblurring) on ImageNet and CIFAR-10, measured by LPIPS and CMMD.

**Strengths:**

- **Theoretical Soundness:** Provides the first rigorous guarantees for DPS in mixture models, explaining both reward and realism improvement.
- **Empirical Effectiveness:** Demonstrates strong, consistent improvements across multiple tasks, datasets, and baselines.
- **Clarity:** Exceptionally well-written and easy to follow.
- **Significance:** Offers a practical, low-cost method to enhance existing diffusion-based solvers, especially for highly compressive inverse problems.

**Weaknesses:**

- **Theoretical Scope:** The theoretical guarantees are currently limited to Gaussian mixture models. While insightful, extending them to more complex distributions remains future work.
- **Computational Overhead:** REGUIDANCE doubles the inference time (inversion + DPS-ODE), though the absolute cost (≤7 GPU minutes) is reasonable. A more detailed runtime comparison would be helpful.
- **Initialization Dependency:** The method’s performance depends on the quality of the initial reconstruction. While it boosts weak baselines, poor initializations may limit gains.

**Questions:**

1. The paper shows the space of good latents is disconnected. Could this be exploited to generate diverse solutions, e.g., by sampling multiple latents from a baseline and applying REGUIDANCE?
2. While REGUIDANCE improves sample quality, does it also improve convergence speed or stability compared to running DPS from random initialization?
3. Have you explored adaptive strategies for choosing the guidance strength $\rho$ or time horizon $T$ based on the task or baseline method?

---

> ### Author Response · Authors · 2025-11-20
>
> We thank the reviewer for their carefully thought-out questions and comments!
>
> **"The theoretical guarantees are currently limited to Gaussian mixture models..."**
>
> Our work provides the first end-to-end theoretical characterization of the sampling for guidance on the Gaussian mixture, and we hope the following context will help calibrate the reviewer’s expectations regarding theoretical results in the literature on diffusion guidance, which is largely dominated by heuristic arguments.
>
> Diffusion posterior sampling targets an algorithmic problem which can be **computationally hard even in the average case** (see [1]). As a result, it has been extremely tricky to identify settings where an end-to-end analysis of a heuristic used in practice is feasible. Indeed, current empirical works restrict their theoretical scope to strongly simplified models of the data. We note that **even among theoretical works on diffusion guidance which have appeared at the main ML conferences recently, it is standard to focus on Gaussian mixtures** (see [2] and [3]). In the particular setting of gradient guidance (DPS) we focus on in this submission, ours is the *first end-to-end theoretical guarantee* completely characterizing the sampling dynamics of the *empirically common algorithm of reward gradient steering*, for any model of data.
>
> **Computational Overhead and Runtime Comparisons**
>
> 7 minutes is the cost for ReGuidance using a DPS baseline, which is the most runtime-consuming of our three baselines. In general, if the baseline method takes time B, and there are N diffusion steps, ReGuidance has runtime $B$ + $N \cdot $ Inv + $N \cdot $ ReG, where Inv is the time per inversion step and ReG is the time per ReGuidance step. Empirically (for ImageNet 256x256), Inv ~ $0.06$ s, while ReG ~ $0.18$ s, since the DPS-ODE requires differentiating through the reward. For N = 1000, our runtime simplifies to $B + 4$ minutes, so methods like DDRM ($B = 20$ sec) and DAPS ($B = 1$ min) are also very reasonable at inference given the performance boost. An added runtime of $4$ minutes per sample is certainly justified given the empirical boosts in performance.
>
> We will include this analysis in the final version of our paper!
>
> **"Initialization Dependency: The method’s performance depends on the quality of the initial reconstruction..."**
>
> We would like to point out that empirically, the bar for initial reconstructions are not particularly high (see Fig. 1 second row, Figs. 9/10) to achieve strong gains in performance. Of course, finding even better and diverse initializations for steering remains an exciting direction for future work.
>
> **"Can disconnected latents be exploited for generating diverse solutions...?"**
>
> Yes! We show this in Figure 3 of our paper (bottom row). Using different candidate solutions (from different baselines), ReGuidance can generate diverse and realistic solutions to hard inverse problems.
>
> **"Does ReGuidance also improve convergence speed or stability?**
>
> In terms of trajectory statistics, given a fixed initialization, DPS would have the same convergence/stability characteristics as ReGuidance since ReGuidance uses a DPS-ODE after the inversion step.
>
> **"Have you explored adaptive strategies for choosing the guidance strength $\rho$... ?"**
>
> We did explore adaptive guidance schedules for steering! Our general observations were that adaptive guidance assists with “local” changes in image quality, not changing overall image semantics but improving slight defects (these findings are reflected in other comprehensive studies such as [4]).
>
> In the regime of hard rewards, existing methods have large scale semantic defects that local improvements can’t change, demanding “global” interventions. Our observation is that strong initializations are one such global fix, so we report results with fixed guidance scales to reflect this local/global separation.
>
> ----------------------------------------------------------
> **References**
>
> [1]: Bruna-Han 2024. “Posterior Sampling with Denoising Oracles via Tilted Transport”
>
> [2]: Wu et. al. "Theoretical Insights for Diffusion Guidance" (ICML ‘24)
>
> [3]: Chidambaram et. al. "What does guidance do?" (NeurIPS '24)
>
> [4]: Yehezkel et. al. 2025. "Navigating with Annealing Guidance Scale in Diffusion Space"

---

### Official Review · Reviewer_ZaHb · 2025-10-31

**Soundness:** 3
**Presentation:** 3
**Contribution:** 3
**Rating:** 8
**Confidence:** 3

**Summary:**

This paper introduces ReGuidance, a simple yet effective method for improving diffusion-based inverse problem solvers in highly compressive or weak-reward settings. The key idea is to invert an existing reconstruction into the diffusion latent space via the reverse probability flow ODE, then re-run deterministic diffusion posterior sampling (DPS) initialized at this latent.

**Strengths:**

- The paper clearly articulates a key weakness of current diffusion guidance techniques in settings with limited measurement information or weak reward signals, and introduces a an approach that systematically addresses this issue.
- ReGuidance is conceptually simple yet powerful. It can be applied to any pretrained diffusion model or inverse problem solver without retraining, making it broadly useful.
- The paper is clearly written, with well-organized motivation, method, and theory.

**Weaknesses:**

- ReGuidance always re-samples from one recovered latent, offering limited posterior diversity. Exploring multiple inverted latents or stochastic variants might provide richer solutions.
- The paper would benefit from a comparison or discussion with D-Flow [1], which similarly optimizes the diffusion starting point to improve reconstruction and control.
- Both the reverse ODE and DPS-ODE stages are deterministic with fixed hyperparameters $\rho$ and $T$. It would be good to have ablations or robustness analysis.

[1] Ben-Hamu et al. D-Flow: Differentiating through Flows for Controlled Generation. ICML 2024.

**Questions:**

See weakness.

---

> ### Author Response · Authors · 2025-11-20
>
> We sincerely thank the reviewer for their thoughtful comments!
>
> **"ReGuidance always re-samples from one recovered latent, offering limited posterior diversity..."**
>
> One way to explore different initializations is to use different baseline solvers (i.e. DAPS/DDRM/DPS), where ReGuidance can capture multiple inverted latents that all lead to diverse but realistic solutions (see bottom row of Figure 3). Examining stochastic alternatives offers a promising avenue to unlock even more sample diversity.
>
>
> **"The paper would benefit from a comparison or discussion with D-Flow [1]..."**
>
> We thank the reviewer for bringing this paper to our attention! Similar to our work, D-Flow observes that certain initializations can lead the generation process to optimize for a particular reward. Crucially, D-Flow uses the pretrained flow/diffusion model, while we use a gradient-guided flow at inference-time with the DPS-ODE. To ensure realistic images, D-Flow regularizes the optimization by ensuring the optimized initialization lies near a Gaussian shell. In contrast, we find that our strong initializations (for the DPS-ODE) do not necessarily lie on such a shell due to the gradient guidance term in the ODE.
>
> **On ablations and robustness.**
>
> Similar to the original DPS paper (Appendix C [1]) we find that within an interval ($\pm 0.2$) of our guidance scales, performance differences are roughly negligible (on average, LPIPS and CMMD vary on the order of $0.01$). Setting the guidance scale too high introduces image artifacts such as grated patterns or oversaturation, and setting the scale too low recreates the candidate solution (i.e. reproduces the inverted baseline image). We will add this discussion to the final version of the paper.
>
> ---------------------------------------------------------
> **References**
>
> [1]: Chung et. al. 2023. Diffusion Posterior Sampling for General Noisy Inverse Problems

---

### Author Response · Authors · 2025-12-02
**Summary of Responses for the AC's Reference**

We thank all the reviewers for their engagement and thoughtful questions during the review process. We are glad that our reviewers have pointed out our "simple yet powerful" (**ZaHb**) technique that is "theoretically sound" (**Xb4y**) and offers "strong, consistent empirical improvements" (**Xb4y**). We also appreciate Reviewer **ev17** pointing out again the "simple effectiveness" (**ev17**) of our strong empirical results as well as commending our focus on "hard datasets" (**ev17**) and "hard inverse problems" (**ev17**) that "have not been paid attention to in the literature" (**ev17**).

We provide a summary of our responses to the reviewers’ major questions below. While we did not get any responses to our rebuttals prior to the discussion freeze, we believe our additional experiments and discussion clearly and comprehensively address the questions that were raised in their initial reviews.

**Experiments: Comparison to Recent Methods**

We added new comparisons to recent MAP (maximum a-posteriori) methods for image restoration to: a) provide additional evidence that ReGuidance is superior to existing methods when it comes to hard reward regimes, and b), emphasize that the novelty and success of our algorithm owes to our novel combination of “strong initializations” *in tandem* with our deterministic guidance process.

The results, included in the thread below, investigate a recent MAP method called DMPlug, and we show that 1) ReGuidance on any of our baselines strongly outperforms DMPlug on its own and 2) ReGuidance offers a boost on top of DMPlug as well, which should entirely resolve Reviewer **ev17**’s primary concerns. Moreover, this should resolve Reviewer **EPZz**'s request for a more modern baseline.

In addition, Reviewer **EPZz** asks for non-ImageNet datasets as well; we point towards our experiments on CIFAR in Appendix E.1 that show our strong improvements transfer to another "hard dataset" (**ev17**).

**Ablations on Initializations**

Both Reviewer **EPZz** and Reviewer **ev17** ask about theoretical and empirical notions of "good initializations". Our theorems imply that discretization errors in inversion can be tolerated, since the condition is a distance function in noise space. As our results demonstrate, the bar for initial reconstruction quality is not quite high either (see Fig. 1, 9, 10), so a "good initialization" while necessary tolerates lower quality baselines.

**Theoretical Scope.**

Our theoretical results offer an exceptionally strong result, especially when juxtaposed with the existing literature. Prior works to ours that offer theoretical results on diffusion guidance rely on overly strong assumptions on the reverse diffusion process that fail to hold even for basic distributions like Gaussian data. For example, [1] assumes access to a posterior distribution that is in general provably intractable to sample from, while [2] assumes the reverse process always stays within the noised data manifold, which holds neither in theory nor in practice.

Instead of making such assumptions, our theoretical results directly analyze the dynamics of the DPS reverse process from start to end; to our knowledge, this is the first analysis of DPS of its kind in the literature. This has proven difficult for general data distributions; indeed, algorithmically, diffusion guidance to a reward is provably NP-hard [3]. We prove the next strongest objective: that reward and realism both strictly improve, and we do this for the only data distribution that has yielded guarantees for guided diffusion to date: mixtures of Gaussians (see also the work of Wu et al. [ICML ‘24] and Chidambaram et al. [NeurIPS ‘24]).

This additional context should resolve Reviewer **EPZz**'s and Reviewer **Xb4y**'s about theoretical strength.

**Ablations for Hyperparameters and Runtime Analysis.**

We include further discussion on choice of hyperparameters and a detailed runtime analysis as requested. We find that, especially compared to MAP methods that seem to have the next best promise for our regime of hard rewards, ReGuidance is an order of magnitude faster (<=7 min per sample compared to 40 min per sample for DMPlug, e.g.). The compute overhead is well justified given our strongly superior results on inpainting, deblurring, and superresolution in the paper. We include more details in the follow-up comment that should resolve Reviewer **EPZz**'s, Reviewer **Xb4y**'s, and Reviewer **ev17**'s request for more information about ablations and runtime analysis.

Again, further details on these points are included in the follow-up comment per the AC’s reference.


-------------------------------------------------
**References**

[1]: Zhang et. al. "Improving Diffusion Inverse Problem Solving with Decoupled Noise Annealing" (CVPR 2025)

[2]: Yang et. al. "Guidance with Spherical Gaussian Constraint for Conditional Diffusion" (ICML 2024)

[3]: Gupta et. al. "Diffusion Posterior Sampling is Computationally Intractable"

---

> ### Author Response · Authors · 2025-12-02
> **Follow-up Details for the AC's Reference**
>
> **Additional Experiments (Comparisons to More Modern/MAP Methods)**
>
> We argue that encouraging MAP is not doing the “heavy lifting” in our empirical results, but it is precisely the strong initializations coupled with the DPS-ODE jointly that lead to performant results. We use the deterministic version of DPS precisely to utilize the initialization information, as an SDE would destroy this information content. As our theorems demonstrate, gradient guidance (DPS) coupled with this initialization improves the realism/reward.
>
> To ablate for this, as the reviewer has suggested, we explore a MAP solver (DMPlug) as a baseline, which simply encourages MAP solutions irrespective of initialization. We found that DMPlug does not perform very well on the large inpainting task (worse than even DAPS without ReGuidance). The DMPlug per sample runtime for ImageNet was $40$ min per sample, compared to our runtime of $\leq 7$ min per sample.
>
> In fact, ReGuidance offers a performance boost on top of DMPlug, but this does not perform nearly as well as DMPlug+DAPS or DMPlug+DDRM, demonstrating that better initializations are crucial to the overall performance of ReGuidance, not just the MAP-tendencies of the DPS-ODE.
>
> | **Method**               | **LPIPS ↓** | **CMMD ↓** |
> |--------------------------|-------------|-------------|
> | DMPlug                   | 0.542       | 1.430       |
> | DMPlug + ReGuidance     | **0.443**       | **1.352**       |
> | DAPS                     | 0.418       | 1.103       |
> | DAPS + ReGuidance       | **0.376**       | **0.628**       |
>
>
> **Ablations on Initializations**
>
> Reviewer **ev17** asks for what is theoretically meant by a "good initialization" and whether intermediate initializations are effective empirically (Q4/Q5/Q6). We use “sufficiently close” as convenient language to express that a "good initialization" is within some distance of a ground truth latent, but the actual theorem does not actually demand very close proximity in initialization, as the experiments demonstrate.
>
> As for intermediate initializations (start ReGuidance at intermediate instead of full inversions), we find that even at a small scale, the artifacts introduced deprecate the image semantics by a strong amount, which (in the official rebuttal) we attribute to a mismatch between the baseline semantics and the reward guidance.
>
>
> **Theoretical Scope.**
>
> We hope the following context will help calibrate the reviewer’s expectations regarding theoretical results in the literature on diffusion guidance, which is largely dominated by heuristic arguments.
>
> Diffusion posterior sampling targets an algorithmic problem which can be **computationally hard even in the average case** (see [1]). As a result, it has been extremely tricky to identify settings where an end-to-end analysis of a heuristic used in practice is feasible. Indeed, current empirical works restrict their theoretical scope to strongly simplified models of the data. We note that **even among theoretical works on diffusion guidance which have appeared at the main ML conferences recently, it is standard to focus on Gaussian mixtures** (see [2] and [3]). In the particular setting of gradient guidance (DPS) we focus on in this submission, ours is the *first end-to-end theoretical guarantee* completely characterizing the sampling dynamics of the *empirically common algorithm of reward gradient steering*, for any model of data.
>
>
> **Ablations for Hyperparameters and Runtime Analysis.**
>
> In terms of ablations, similar to the original DPS paper (Appendix C [4]) we find that within an interval ($\pm 0.2$) of our guidance scales, performance differences are roughly negligible (on average, LPIPS and CMMD vary on the order of $0.01$). Setting the guidance scale too high introduces image artifacts such as grated patterns or oversaturation, and setting the scale too low recreates the candidate solution (i.e. reproduces the inverted baseline image).
>
> In general, if the baseline method takes time $B$, and there are $N$ diffusion steps, the total time for ReGuidance is $B$ + $N \cdot$ Inv + $N \cdot$ ReG, where Inv is the time per inversion step and ReG is the time per ReGuidance step. Empirically (for ImageNet 256x256), Inv ~ $0.06$ s, while ReG ~ $0.18$ s, since the DPS-ODE requires differentiating through the reward. For $N = 1000$, the ReGuidance has runtime $B + 4$ min, so methods like DDRM ($B = 20$ sec) and DAPS ($B = 1$ min) are reasonable at inference given the performance boost. In comparison, a MAP method like DMPlug ends up taking $40$ min per sample for ImageNet 256x256.
>
> -----------------------------------------------------------------
> **References**
>
> [1]: Bruna-Han 2024. “Posterior Sampling with Denoising Oracles via Tilted Transport”
>
> [2]: Wu et. al. "Theoretical Insights for Diffusion Guidance" (ICML ‘24)
>
> [3]: Chidambaram et. al. "What does guidance do?" (NeurIPS '24)
>
> [4]: Chung et. al. 2023. "Diffusion Posterior Sampling for General Noisy Inverse Problems"

---

### Meta-Review · Area_Chair_si2V · 2026-01-06

**Summary:**

The paper proposes ReGuidance: it takes an existing reconstruction from a diffusion-based inverse solver, inverts it into latent space via the reverse probability flow ODE, and then runs deterministic DPS-ODE starting from that latent to improve performance on hard inverse problems (high compression / weak guidance). The idea is simple and modular, and the empirical section reports strong improvements on several challenging settings. The decision, however, is driven by whether this is a sufficiently substantive contribution beyond a well-engineered refinement of existing DPS-style pipelines.

**Reviewer Concerns:**

Concerns addressed by the rebuttal:
1) Missing baselines and “is this just MAP?”: the rebuttal adds a DMPlug comparison and argues the benefit is not only MAP-like optimization, but the combination of strong latent initialization and deterministic guidance.
2) Practical details: runtime reporting and some ablation/robustness clarifications were added, reducing uncertainty about implementation choices and overhead.
3) Evaluation protocol questions (e.g., sample size): the rebuttal provides an expanded evaluation check and argues the trend is stable.

Concerns still outstanding:
1) Novelty/impact remains the central issue. Even with improved positioning, the method still reads as a particular recipe for better initialization + deterministic steering within an existing framework, rather than a new insight into posterior sampling or guidance.
2) Theoretical support is limited to simplified settings and does not meaningfully explain why/when the method should work across the broad set of empirical claims.
3) The deterministic “single-latent” nature raises questions about posterior diversity and whether the method is best viewed as improving point estimates rather than posterior sampling; this is discussed but not convincingly resolved.

**Reviewer Scores:**

1) Reviewer (score 8): likely unchanged at 8; the rebuttal reinforces their positive view and addresses several practical points they cared about.
2) Reviewer (score 4): likely unchanged at 4. The added comparisons/runtime details are helpful, but their core hesitation is novelty/impact, and the rebuttal does not substantially change that.
3) Reviewer (score 4): likely unchanged at 4 for similar reasons; the response reduces uncertainty on implementation details but does not move the needle on conceptual contribution.
4) Reviewer (score 4): likely increase to 5. This reviewer’s concerns were more about missing baselines/ablations and evaluation protocol, and the rebuttal makes a meaningful improvement there.

---

### Decision · Program_Chairs · 2026-01-26

Reject